# TEXT-AWARE DIFFUSION POLICIES

## ABSTRACT

Diffusion models scaled to massive datasets have demonstrated powerful underlying unification capabilities between the language modality and pixel space, as convincingly evidenced by high-quality text-to-image synthesis that delight and astound. In this work, we interpret agents interacting within a visual reinforcement learning setting as trainable video renderers, where the output video is simply frames stitched together across sequential timesteps. Then, we propose Text-Aware Diffusion Policies (TADPols), which uses large-scale pretrained models, particularly text-to-image diffusion models, to train policies that are aligned with natural language text inputs. As the behavior represented within a policy naturally learns to align with the reward function utilized during optimization, we propose generating the reward signal for a reinforcement learning agent as the similarity between a provided text description and the frames the agent produces from its interactions. Furthermore, rendering the video produced by an agent during inference can be treated as a form of text-to-video generation, where the video has the added bonus of always being smooth and consistent with respect to the environmental specifications. Additionally, when the diffusion model is kept frozen, this enables the investigation of how well a large-scale model pretrained only on static image and textual data is able to understand temporally extended behaviors and actions. We conduct experiments on a variety of locomotion experiments across multiple subjects, and demonstrate that agents can be trained using the unified understanding of vision and language captured within large-scale pretrained diffusion models to not only synthesize videos that correspond with provided text, but also learn to perform the motion itself as autonomous agents.

## 1 INTRODUCTION

In reinforcement learning, agents learn to perform a behavior as specified by a reward function provided during training (Sutton & Barto, 2018). The reliance on manually designed reward functions naturally renders inscalable the learning of ever-increasing amounts of novel behaviors; having to hand-create reward functions for novel behaviors is a naturally intractable endeavour. Indeed, the design of each reward function requires a certain level of care, as naive reward functions can produce agents that are prone to reward hacking (Skalse et al., 2022). As a result, for many environments it is common to have only one or a few reward functions provided, which are carefully defined to produce specific desired behaviors. For example, the only reward functions associated with the Dog environment from the DeepMind Control Suite (Tunyasuvunakool et al., 2020) are constructed to promote fetching and various speeds of walking behavior. If we also wished to learn a Dog agent that can roll on the ground, or jump into the air, we would have to carefully design their respective reward signals ad hoc. Instead, it is highly desirable to be able to train an agent to perform a desired behavior specified directly through a convenient and general interface such as natural language, as language can easily and flexibly describe behaviors of interest. How to construct an accurate and expressive reward signal for a reinforcement learning agent as a function of natural language specifications is therefore a worthwhile goal.

Another bottleneck for advancement in reinforcement learning stems from its lack of access to large-scale data. Exciting developments in natural language modeling, vision, and even their intersection have been closely tied to a ballooning in architecture size (Kaplan et al., 2020; Zhai et al., 2022), and crucially, an ever-increasing scale of observed data during optimization (Sun et al., 2017). Recently, imitation learning (Torabi et al., 2018; Lynch et al., 2023) has gathered interest as a way to introduce

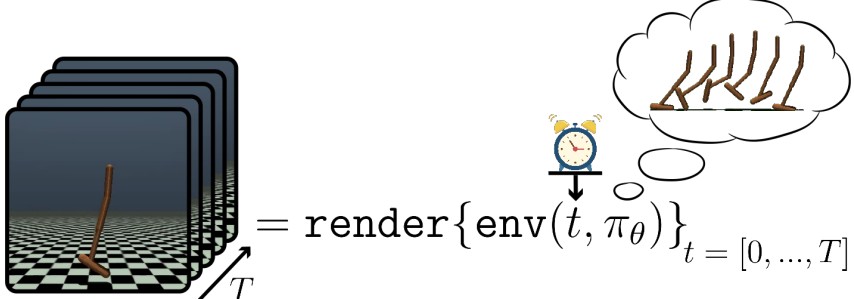

$$= \texttt{render}\{\texttt{env}(t, \pi_\theta)\}_{t = [0, ..., T]}$$

Figure 1: A policy $\pi_\theta$ that interacts with an environment can be treated as an implicit video representation network, denoted here visually as "dreaming" of the behavior it encodes. At every timestep $t$, the environment updates according to the behavior defined by the policy; by rendering each sequential state over multiple timesteps $t = [0, ..., T]$ and stitching them together, the implicit video representation captured within a policy is realized as an actual video.

scale to the learning of behaviors when a large amount of expert demonstrations are available. However, such demonstrations may be expensive or difficult to generate, and in these cases where access to an expert dataset is unavailable, reinforcement learning is generally applied. Without access to internet-level amounts of data, traditional reinforcement learning has therefore been largely isolated from the exciting developments of scale as observed for text or vision. Reinforcement learning networks and data design has mostly stagnated in recent years, with researchers generally reusing the same limited architecture sizes, and treating the same environments and/or offline trajectories as data sources. How to integrate large-scale models into reinforcement learning agents, as well as utilize priors extracted from internet-level data effectively for optimization, is therefore of great interest.

In this work we present Text-Aware Diffusion Policies (TADPols), which seeks to address both shortcomings of existing reinforcement learning agents simultaneously; we propose crafting a general-purpose reward function that is conditioned flexibly on natural language, and provides expressive rewards to the agent leveraging priors from internet-scale data, by utilizing pretrained, frozen text-to-image diffusion models (Dhariwal & Nichol, 2021; Ramesh et al., 2022; Saharia et al., 2022; Rombach et al., 2022).

Diffusion models (Sohl-Dickstein et al., 2015; Song & Ermon, 2019; Ho et al., 2020; Song et al., 2020b) have recently demonstrated amazing generative modeling capabilities, particularly in the domain of text-conditioned image generation (Dhariwal & Nichol, 2021; Ramesh et al., 2022; Saharia et al., 2022; Rombach et al., 2022). Notably, guidance (Song et al., 2020b; Dhariwal & Nichol, 2021; Ho & Salimans, 2022) has been shown to be a critical component in producing visual outputs aligned with textual data, enabling the generation of images that accurately match a desired text caption not only in terms of depicted subjects, but also stylistically and aesthetically. When scaled to utilize large foundation models such as CLIP (Radford et al., 2021) or T5 (Raffel et al., 2020), and trained on massive image-text datasets (Schuhmann et al., 2021), the resulting diffusion models have demonstrated powerful generative capabilities that suggest an underlying deep unified understanding of image and text.

The key insight that reveals the natural connection between reinforcement learning agents and text-to-image diffusion models is that a policy can be treated as an implicit video representation network when operating within an environment with visual rendering capabilities. The video described within the policy can be visualized through the built-in rendering function of the environment, by repeatedly rendering at each step of interaction, as depicted in Figure 1. By treating the policy as an iterative frame renderer that generates coherent, reward-aligned images through the actions it selects, we can draw a parallel with text-to-image diffusion models, which also generate coherent images, which are instead text-aligned. This shared perspective inspires the learning of text-conditioned policies by using a generative diffusion model in a discriminative manner to provide text-conditioned rewards. In this way, we effectively distill the priors and text-alignment captured within a pre-trained text-to-image diffusion model such that the resulting policy also selects actions to generate frames that are now strongly aligned with human-provided text.

Through TADPol, the benefits of large-scale pretrained data and models can be realized in the reinforcement learning framework. We showcase these immediate scaling properties by demonstrating that TADPol enables the learning of novel, zero-shot policies that are flexibly and accurately conditioned on natural language inputs, across multiple robot configurations and environments. On the complex Dog environment from the DeepMind Control Suite (Tunyasuvunakool et al., 2020), we demonstrate that TADPol can learn novel goal-achieving behaviors (such as standing on its hind legs or chasing its tail) as well as continuous locomotion (such as walking) from text-conditioning alone. Furthermore, as a trained policy can be interpreted as a reward-aligned implicit video representation, optimizing via TADPol can also be thought of as learning a text-aligned video representation. Then, text-to-video generation can be simulated by stitching together sequential renderings from the environment as a trained TADPol agent performs inference. We note that these generated text-conditioned videos are naturally limited by the inherent visual qualities of the environment the agent interacts with. However, as they were trained in alignment with the priors of a frozen text-to-image diffusion model, these visualizations can provide initial interesting insights into how well such text-to-image diffusion models understand temporally-extended behavior and verbs, despite having only been trained on natural language and static images.

## 2 BACKGROUND AND RELATED WORK

Recent work has studied whether pretrained, frozen, text-to-image diffusion models naturally encode and understand 3D shape information. In other words, despite being trained only from static 2D images alongside text captions, a large-scale diffusion model may understand when an image depicts an object from an angle, such as from the rear or the side, and therefore can potentially be used as a supervisory signal to learn a 3D volume such as a Neural Radiance Field (NERF) (Mildenhall et al., 2020). This is at the heart of the DreamFusion model (Poole et al., 2022), which uses a pretrained, frozen ImageN (Saharia et al., 2022), and optimizes a NERF through their proposed Score Distillation Sampling technique. At a high level, this result suggests that powerful, large-scale pretrained image-to-text diffusion models exhibit emergent understanding beyond the 2D images and textual data they were trained on into higher dimensionalities such as volume. Taking inspiration from this work, we attempt to interface with frozen, pretrained diffusion models in a similar way, and investigate if such models exhibit emergent understanding into the temporal dimension.

Prior work on video diffusion models already demonstrate that text-to-image models can generate still images that represent verb terms, and that with some simple modifications, they are able to generate multiple images concurrently that exhibit content consistency (Khachatryan et al., 2023; Wu et al., 2022). Tune-A-Video takes in a text-video pair and finetunes a pretrained text-to-image diffusion model to reconstruct it; then, a novel video can be generated by inverting the original input video into noise, and guiding generation through the finetuned model with an edited prompt. Imagen Video utilizes a low-resolution base video generation model in conjunction with cascaded spatial and temporal super-resolution models to generate HD videos from text (Ho et al., 2022). Make-A-Video decodes an image embedding produced from a pretrained text-to-image model into multiple frames, which are progressively interpolated to higher frame rates and spatial resolutions, to ultimately produce a high-resolution video conditioned on the original text. (Singer et al., 2022). Rather than corrupting frames of a video independently during generation as in standard diffusion, VideoFusion proposes a decomposed diffusion process where per-frame noise is decomposed into a consistent, shared base noise and a residual noise that varies across time, for more performant text-to-video generation (Luo et al., 2023). In this work, we treat an agent that interacts with an environment with visual rendering capabilities as an implicit video representation network. Then, for an agent well-trained to align with a text caption, an associated video can be rendered as it interacts with the environment.

Prior art has investigated how diffusion can power the learning of reinforcement learning agents. Janner et al. (2022) propose treating planning as an iterative refining trajectories using a denoising procedure. Diffusion-QL implements the policy directly as a conditional diffusion model to solve offline reinforcement learning tasks (Wang et al., 2022). In this work, we leverage pretrained, frozen diffusion models to learn powerful agents aligned with natural language text specifications.

There are also numerous works that investigate how interactive agents can learn to perform behaviors specified by textual inputs. SayCan (Ahn et al., 2022) grounds the knowledge of complex, high-level

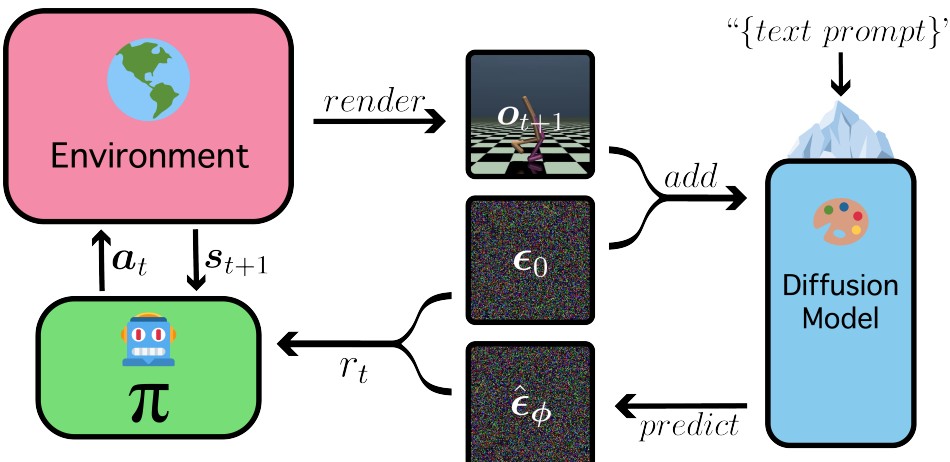

Figure 2: An illustration of our proposed pipeline. TADPol is composed of a policy that interacts with an environment with rendering capabilities and a frozen, pretrained, text-to-image diffusion model. At each timestep, the subsequent frame rendered by the environment is converted to a reward signal as an alignment score with respect to a provided text prompt. The rendered image from the environment is first corrupted with a sampled Gaussian source noise vector, and then the diffusion model is utilized to predict the source noise that was added. The reward signal is then computed as a function of the noise prediction error; if the text prompt is well-aligned, then the noise prediction error will be small and a larger positive reward will be assigned to the agent.

behaviors within an LLM to the context of a robot through pretrained behaviors. This then enables a LLM to instruct and guide a robot, through combining low-level behaviors, to perform complex temporally extended behaviors. LangLfP proposes a method for incorporating free-form natural language conditioning into imitation learning by first associating goal images with text captions and training a policy to follow either language or image goals, but only conditioning on natural language during test time inference (Lynch & Sermanet, 2020). The Text-Conditioned Decision Transformer learns a causal transformer to autoregressively produce actions conditioned on both text tokens as well as state and action tokens (Putterman et al., 2021). Similarly, Hiveformer proposes a unified multimodal Transformer model for robotic manipulation that conditions on natural language instructions, camera views, as well as past actions and observations (Guhur et al., 2023). Hierarchical Abstraction with Language (HAL) is a hierarchical policy where a high-level policy produces compositional natural language instructions to interface with a low-level policy(Jiang et al., 2019). SuccessVQA treats the detection of successful behavior as a visual question answering task, and develop success detectors using large, pretrained vision-language models; such an approach provides a way to convert natural language and observed visual data into reward signals with which to learn intelligent agents (Du et al., 2023). LIV learns a vision-language representation that implicitly encodes a value function for tasks specified by language and image goals, which can then be used to specify zero shot reward values for novel videos (Ma et al., 2023).

Similar to our work, Dai et al. (2023) treat the sequential decision-making problem as a text-conditioned video generation problem, and seek to leverage the capabilities of generative models to learn performant policies. The authors propose training a video diffusion model to produce a future visual plan for the agent; the subsequent frames are then converted to actions by means of an inverse dynamics model. In our work, however, we directly use large-scale, frozen pretrained text-to-image models that have not observed video data. Also similar to our work, Mahmoudieh et al. (2022) propose a framework that uses CLIP to generate a reward signal from a text description of a goal state and raw pixel observations from the environment, which is then used to learn a task policy. In our work, we focus on investigating locomotion tasks rather than goal-achievement, and explore how using diffusion models to produce a reward signal fundamentally outperforms CLIP.

## 3 METHOD

By nature, a policy implicitly encodes a behavior with respect to a reward function provided during training, where the behavior captured in the weights of the policy network can be instantiated in the

form of a trajectory $\tau = (s_0, a_0, r_0, s_1, a_1, ...)$ during inference with respect to the environment the agent interacts with. With respect to environments with visual rendering capabilities, however, a policy can also be treated as an implicit video representation network. As the agent interacts with such an environment during inference, a trajectory $\tau_{obs} = (s_0, o_0, a_0, r_0, s_1, o_1, a_1, ...)$ is realized. As sequentially stitching together all $v_\pi = \texttt{concat}([o_0, o_1, ...])$ produces a video, the trajectory determined by a policy can therefore be interpreted as an annotated video, where each arbitrary frame $o_t$ is additionally supplied with action $a_{t-1}$, reward $r_t$, and optionally, state $s_t$ (where in pixel-based control, $s_t$ is equivalent to $o_t$). Leveraging the environment's rendering function $P(o_t \mid s_t)$, a policy can therefore be interpreted as an optimizable implicit video representation network, where the video it learns to encode with respect to the environment is trained to align with a provided reward function. For example, if a policy is optimized with respect to a reward function that encourages the behavior of running, a well-trained policy would then produce a video of the agent running when sequentially rendered through the environment during inference. As the reward function shapes the implicit video representation of a policy, a natural question of interest is how to define the reward function with respect to a natural language text input. In doing so, the ability to train a text-to-video model as a reinforcement learning agent in a flexible visual environment is unlocked; furthermore, the resulting video carries the additional guarantee of smoothness and physical consistency with respect to adjustable parameters that underpin the environment. Crafting an expressive reward function that accurately conditions on natural language also enables the learning of interesting policies guided not through carefully hand-designed ad-hoc scalar manipulations, but through the more natural interface that is human language.

In this work we propose Text-Aware Diffusion Policies (TADPols), where the behavior and video implicitly specified by a reinforcement learning agent is trained to align with natural language text captions by leveraging a frozen, pretrained diffusion model. This is motivated by the intuition that if the agent takes an action that produces a rendered image that aligns well with the given text caption, as judged by a large-scale pretrained model that unifies pixel and text modalities, then the action should be rewarded highly. When leveraging a text-to-image model (Ramesh et al., 2022; Saharia et al., 2022; Rombach et al., 2022), the prior belief we operate on is that a generated video is overall well-aligned with a desired text prompt if each frame of the video is independently well-aligned with the text prompt. In this work, we focus our efforts on utilizing a text-to-image model, but also consider the benefits of using pretrained text-to-video diffusion models in our experiments. The large-scale nature of the text-to-image models utilized as a reward signal enables the zero-shot learning of policies as specified by a natural language input, across many potential visual domains and environments. Furthermore, keeping the pretrained models frozen affords us the ability to preliminarily investigate how well such models naturally understand verbs and temporally extended behavior, despite only having been trained on static images and textual data. Such insights can be gleaned from analyzing how well the resulting policy aligns with the provided text prompt, such as through visualizing the generated videos from inferred trajectories.

At each timestep $t$, reward $r_t$ is computed as the alignment between rendered subsequent image $o_{t+1}$ and the provided text caption describing the behavior of interest. The alignment score is computed from a frozen, pretrained StableDiffusion (Rombach et al., 2022) checkpoint through a modified version of score distillation sampling (Poole et al., 2022) as described below. We corrupt the rendered image $\mathbf{o}_{t+1}$ with a sampled Gaussian source noise vector $\epsilon_0$ to produce noisy observation $\tilde{\mathbf{o}}_{t+1}$, and use a frozen, pretrained diffusion model to predict the source noise as $\hat{\epsilon}_\phi(\tilde{\mathbf{o}}_{t+1}; t_{\text{noise}}, y)$ where $t_{\text{noise}}$ denotes the level of noisy corruption and $y$ denotes the provided text caption. The mean squared error between the source noise and the prediction is then converted into a scalar reward signal at each time step representing the similarity with the provided text caption:

$$r_t = \exp\left(-\kappa(t_{\text{noise}}) \left\| \epsilon_0 - \hat{\epsilon}_\phi(\tilde{\mathbf{o}}_{t+1}; t_{\text{noise}}, y) \right\|_2^2\right)$$

It is worth emphasizing that $t_{\text{noise}}$ and subscript-less $t$ refer to different notions of time; $t$ indexes the timestep of the RL agent in the environment, whereas $t_{\text{noise}}$ determines the level of noise to corrupt the raw observed image. The exponent of the negative mean squared error is used to bound the reward value; as the mean squared error diminishes, the reward signal approaches 1; conversely, as the mean squared error explodes, the reward signal approaches 0. Additional hyperparameters to control the variance of the resulting reward signal, optionally factoring in the noise level $t_{\text{noise}}$, can be flexibly tuned. Additional multiplicative or additive scales to further shape $r_t$ can be considered as well. This framework enables the definition of a dense reward signal, leveraging only a prior pretrained text-to-image diffusion model, and the natural rendering capability of the environment.

**Algorithm 1** Text-Aware Diffusion Policy (TADPol)

```
 1: txt_caption = sample(action_phrase)
 2: π_θ = initialize(θ)
 3: D ← {}
 4: while not converged:
 5:     s_0 ∼ p(s_0)
 6:     for t in range(episode_length):
 7:         a_t ∼ π_θ(a_t | s_t)
 8:         s_{t+1} ∼ P(s_{t+1} | s_t, a_t)
 9:         o_{t+1} ∼ P(o_{t+1} | s_{t+1})
10:         õ_{t+1} ← noisify(o_{t+1}, t_noise)
11:         r_t = exp(−κ(t_noise) ‖ε_0 − ε̂_φ(õ_{t+1}; t_noise, txt_caption)‖²₂)
12:         τ ← τ ∪ (s_t, a_t, r_t, s_{t+1})
13:     D ← D ∪ τ
14:     loss = policy_loss(D)
15:     grads = gradient(loss, θ)
16:     opt.apply_gradients(grads, θ)
```

TADPol is agnostic to the the choice of policy network architecture and optimization; indeed, TADPol can be flexibly applied to a variety of existing policy networks and objectives to provide natural language conditioning. A pseudocode of the method is provided in Algorithm 1.

## 4 EXPERIMENTS

We conduct thorough experimentation to investigate how TADPol successfully enables policies to learn behaviors aligned with human-specified text prompts. We compare the resulting policies with those trained from ground-truth specified reward functions, and show that decent performance is achieved purely from text conditioning. We further investigate the choice of using a text-to-image diffusion model as the alignment component in TADPol against other alignment models, such as CLIP (Radford et al., 2021) and video diffusion (Luo et al., 2023).

In all experiments, the underlying policy is an Actor-Critic model (Barto et al., 1983; Mnih et al., 2016) trained using Robust Policy Optimization (Rahman & Xue, 2022) with the Adam optimizer (Kingma & Ba, 2014). TADPol builds off the base RPO implementation from CleanRL (Huang et al., 2022), and makes no modifications to the standard network architecture and most hyperparameters, apart from using 1 million total steps per task (as opposed to 8 million). We demonstrate our results on a variety of environments, particularly locomotion, from OpenAI Gym (Brockman et al., 2016). Locomotion, in contrast to achieving goal states (Mendonca et al., 2021), is an interesting task with which to evaluate how well pretrained large-scale text-to-image models, as well as agents trained through them via TADPol, innately understand verbs and continuous motion, as no singular frame alone captures the essence of the verb. Rather than learning to search for a goal state regardless of the path, in locomotion, the agent must demonstrate over multiple timesteps a consistent behavior that is aligned with the provided text prompt describing the motion. Learning locomotion and continuous control behaviors from natural language specifications have therefore been a difficult, but interesting task to tackle.

In DreamFusion, the noise level is resampled from a predefined range for every render query during optimization. In TADPol, we find that sampling from a large range results in a reward signal with very high variance. Furthermore, additional variance is introduced when re-sampling the source noise vector at each interaction timestep. As high-variance reward signals can cause instability in training, we reduce the variance of our TADPol reward signal by first maintaining a consistent noise level across all trajectories, and maintaining a consistent source noise that is re-sampled at the beginning of each episode. For the experiments below, we utilize a noise step of 450 out of 1000, where the noise coeffient follows the DDIM Scheduler (Song et al., 2020a). We demonstrate that TADPol is able to synthesize behaviors that achieve convincing quantitative metrics, as well as videos that are qualitatively satisfying with respect to natural language text inputs.

Table 1: A comparison of achieved ground-truth environmental returns across different reward signals. Note that TADPol is not trained on ground-truth environmental signals, but manages to achieve high ground-truth returns despite being optimized purely to align with a provided text prompt.

| Environment | Environment Reward | TADPol |
|---|---|---|
| Hopper-v3 | 3852.78 | 2925.56 |
| Walker2d-v4 | 3872.541 | 2038.06 |
| InvertedDoublePendulum-v4 | 1166.38 | 1474.22 |

Table 2: A comparison of achieved distances and velocities on Walker2d-v4 between CLIPSimPol, TADPol, and Vid-TADPol. We discover that compared to CLIP, diffusion-based models provide a better reward signal to supervise the learning of motion behavior in policies.

| Method | Final $x$-axis Distance | Total Velocity |
|---|---|---|
| CLIPSimPol | 6.259 | 854.324 |
| TADPol | 8.167 | 1083.69 |
| Vid-TADPol | 8.096 | 1060.71 |

## 4.1 Defined Tasks

By nature, TADPol enables the learning of behaviors conditioned on flexible, arbitrary text prompts. However, in the pursuit of quantitative evaluation, we utilize prompts that recreate the behaviors already specified by existing, hand-designed reward functions. By evaluating the behavior of the policy trained by TADPol in terms of the ground-truth reward function, we can benchmark the performance of our approach. We report our results across a variety of experiments from OpenAI Gym in Table 1. The prompts used for each environment were "a one-legged stick figure jumping forward over a grid floor", "a stick figure walking across a grid floor", and "a stick balancing vertically" for the Hopper-v3, Walker2d-v4, and the InvertedDoublePendulum-v4 tasks, respectively. Note that these prompts were not selected from an exhaustive search, but were simply the result of a human attempt to verbally describe the intended behavior promoted by the ground-truth reward function; there is opportunity for prompt tuning to produce additional performance benefits.

We first observe that the rewards achieved for Hopper-v3 and InvertedDoublePendulum-v4 tasks are decent compared to a model trained on the ground-truth reward signal; considering that TADPol does not access the ground-truth reward signal during training, this suggests that TADPol is powerful and flexible enough to simulate existing reward functions with an appropriately-selected text prompt. Furthermore, investigating the training curves of the achieved ground-truth reward signal reveals similar trends between TADPol and a vanilla-trained agent, suggesting that training dynamics are roughly preserved, and that the TADPol reward is a stable feedback signal for optimization.

We then investigate the qualitative results of the videos produced by TADPol against a vanilla agent. We observe that beyond being able to perform actions such as hopping and walking, the resulting videos produced by a trained TADPol agent during inference visually align with human understandings of the respective motions. For example, the walker agents exhibit bending of the knees, and taking well-defined steps. On the other hand, the policy trained using the default reward function simply causes the agent to lean forward and patter its feet to propel it forward, resulting in a rather unnatural motion. We argue that this is because the reward signal, simply catering to low-level scalar values such as speed, is unable to factor in the natural aesthetics of the walking motion. Videos are provided in the supplementary materials, where each task has its own dedicated folder name.

## 4.2 Comparison To CLIP

A natural question to consider is whether models trained directly using alignment objectives such as CLIP (Radford et al., 2021) produce a more suitable alignment score between an image observation and a text caption for the purposes of training a policy aligned with natural language. In this variant,

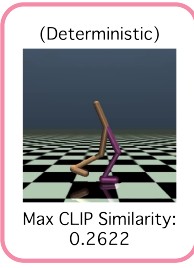
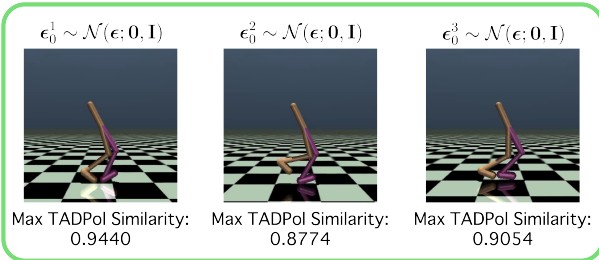

Figure 3: Whereas CLIP deterministically assigns scores to frames from the environment with respect to the text prompt, the reward function calculated via TADPol has different notions of the "most canonical" pose depending on the sampled source noise. This source of stochasticity helps TADPol outperform CLIPSimPol in learning motion policies.

which we term CLIP Similarity Policy (CLIPSimPol), we simply replace the reward function with the cosine similarity as evaluated by a pretrained, frozen CLIP model:

$$r_t = \frac{CLIP_V(\boldsymbol{o}_{t+1})}{\|CLIP_V(\boldsymbol{o}_{t+1})\|} \cdot \frac{CLIP_T(y)}{\|CLIP_T(y)\|}$$

We note that this approach was independently and concurrently proposed by Rocamonde et al. (2023). The CLIP model can be used directly since intuitively, a high alignment score between the achieved observation image and the input text prompt should naturally translate to a higher reward to the agent. We thoroughly investigate the difference between CLIPSimPol and TADPol using the Walker2d-v4 environment. To standardize the comparison, we implement CLIPSimPol and TADPol using the exact same underlying design decisions; all hyperparameters, architectures, and random seeds are kept consistent, and only the reward computation is modified. To further highlight the comparison, we further make adjustments to the underlying reward function. By default, the reward function of Walker2d-v4 rewards the agent for moving to the right; however, the policies learned by CLIPSimPol and TADPol may learn to walk backwards, when provided the prompt "a stick figure walking across a grid floor", which is equally valid as an instantiation of the walking behavior. Furthermore, the environmental reward includes extraneous terms such as a healthiness score and control costs. We therefore compare the two approaches using two metrics: the aggregated velocities, denoting how fast the agent ultimately is able to move, and the final distance achieved by the agent away from its initialization location, which is presented in Table 2.

We observe that CLIPSimPol achieves a noticeably smaller max delta distance along the x-axis from the initial position compared to TADPol, as well as a smaller aggregated velocity in a consistent direction. These results suggest that the CLIP model does not incentivize the agent to learn motion, contrary to our desired goal of learning interesting temporally-extended behaviors. Qualitative inspection of the resulting videos confirm these observations; CLIPSimPol models ultimately learn to converge to a stationary frame, and attempt to stably maintain its position. On the contrary, TADPol learns to continue moving, even after convergence. A proposed explanation for this phenomenon is provided in detail in Appendix A.1, and visualized in Figure 3.

## 4.3 COMPARISON TO VIDEO DIFFUSION

Conceptually, there are fundamental limitations to using a text-to-image model to provide a reward signal. Firstly, as each image is evaluated statically and independently, we are unable to expect the text-to-image diffusion model to be able to accurately understand and supervise an agent's learning of notions of speed, or in some cases, direction, as such concepts require evaluating multiple consecutive timesteps to deduce. For example, from a single snapshot image of a ball in the air, it is difficult to determine how fast it is traveling, or in what direction it is traveling. This fundamental limitation raises the question of whether or not current pre-trained text-to-video diffusion models are more useful in supervising the learning of interesting behaviors in reinforcement learning agents. To evaluate this, we propose Vid-TADPol, where the entire video captured in a trajectory produced by an agent is evaluated using a pretrained, frozen text-to-video diffusion model; in particular, we utilize a VideoFusion (Luo et al., 2023) checkpoint. Dense rewards are then computed for the entire

trajectory through a similar modified score distillation sampling signal as in TADPol:

$$[r_0, r_1, \ldots r_T] = \exp\left(-\kappa(t_{noise}) \left\|[\boldsymbol{\epsilon}_0]_{0\ldots T} - \hat{\boldsymbol{\epsilon}}_\phi(\tilde{\boldsymbol{o}}_{0\ldots T}; t_{noise}, y)\right\|_2^2\right)$$

Here, to clarify any notational confusion, the mean squared error is applied independently for every timestep to generate the reward signal. Furthermore, in our experiments, we allow source noise $[\boldsymbol{\epsilon}_0]_{0\ldots T}$ to vary across timesteps $(0, \ldots T)$. As VideoFusion has observed natural video data during its own training, the resulting policy trained under Vid-TADPol is therefore optimized using knowledge distilled from large-scale natural video datasets. In essence, this allows the agent to leverage priors regarding motion from natural video data to learn similar, text-aligned behavior within the environment it operates in.

We perform comparative experimentation against TADPol, once again using Walker2d-v4 as the environment. Interestingly, however, the results as provided in Table 2 show that Vid-TADPol achieves comparable performance in terms of movement metrics. This could be attributed to a variety of potential factors; for example, perhaps learning to walk in the Walker2d-v4 environment is relatively insensitive to concepts encoded in the text-to-video model beyond a static text-to-image model.

### 4.4 Prompt Sensitivity

An interesting aspect of TADPols to investigate is how sensitive it is to the input prompts, with a particular focus on sensitivity to verbs. To evaluate this, we try three separate prompts on the Hopper-v3 environment: "hopping forward", "jumping up", and "try your best". During training, the best performance achieved by each of the prompts is 2393.78, 2133.22, and 1209.31, respectively. The agent optimized by conditioning on the uninformative prompt learns to keep itself alive, but does not learn any interesting behaviors. On the other hand, we discover that after convergence, the policy trained from the "hopping forward" prompt continues to hop in the forward direction, whereas the "jumping up" prompt produces an agent that hops up and down mostly in place. We take this as evidence that TADPol is sensitive to the input prompts.

### 5 Conclusions and Future Work

We present Text-Aware Diffusion Policies (TADPols), a framework to optimize a reinforcement learning agent to interact with a visual environment in alignment with a provided natural language input. TADPol achieves this by utilizing a pretrained text-to-image diffusion model to sequentially evaluate how aligned the rendered results of an agent's actions are to the desired text caption; these alignment scores are then fed to the agent as dense reward signals to shape its behavior during optimization. The TADPol framework comes with a number of benefits. Firstly, it unlocks the ability to train a policy to exhibit a desired behavior through natural language specification, which is an intuitive and flexible interface for humans to use. Secondly, as an agent can be treated as an implicit video representation, where a video is generated by stitching together sequential renderings from the environment as the agent performs inference, TADPol can essentially perform text-to-video generation where the resulting video is always physically consistent and smooth with respect to parameterizable environment factors (such as physical constants and frame rate). Furthermore, TADPol's usage of pretrained models provides reinforcement learning agents access to large-scale data and priors, and enables them to learn powerful policies across different visual environments and tasks. Lastly, as TADPol uses a frozen text-to-image model, the framework can also provide insight into how well pretraining on static images and textual data encodes temporally extended behavior and verbs. We demonstrate that TADPols are able to successfully learn behaviors aligned with natural language conditioning, and compare against alternative techniques leveraging other foundation models.

A natural limitation of the work is that the videos produced and behaviors learned are directly constrained by the environment the agent interacts with. Future interesting work includes investigating TADPol's ability to learn expressive policies in more powerful and flexible environments, as well as alternative ways to extract the information latent within large-scale pretrained multimodal foundation models for reinforcement learning.

## 6   REPRODUCIBILITY

TADPol is implemented entirely using available open-sourced components. As mentioned in Section 4, TADPol builds off of the base RPO implementation from CleanRL (Huang et al., 2022). Minimal adjustments are made to the codebase, with the exception of modifying the reward signal and rendering functions. Furthermore, in this work we leverage publicly available checkpoints of large-scale foundation models: CLIP (Radford et al., 2021), StableDiffusion (Rombach et al., 2022), and VideoFusion (Luo et al., 2023). We believe that the simplicity of our approach, along with the utilization of open-sourced checkpoints, makes this work highly reproducible.

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

# A  Appendix

## A.1  Investigating the Properties of CLIP and Diffusion as a Reward Signal

The discrepancy outlined in Section 4.2 between the performance of TADPol and CLIPSimPol can be understood by considering the properties of the pretrained CLIP model. CLIP defines a deterministic similarity score between the provided text caption and every possible visual observation determined renderable by the environment. Then, framing the policy as a "search" algorithm that seeks the highest reward, it is naturally incentivized to locate and maintain the position that gives it a consistent high reward from the pretrained CLIP model. Therefore, we can expect the overall distance to be small, provided it identifies a local minimum to exploit, and the overall velocity performed to be small as well, as it tries to stop moving once it achieves a local minimum.

On the other hand, for TADPol, the stochastic nature of how the modified score distillation sampling reward is computed naturally avoids this pitfall. Each stochastic denoising of a corrupted image enables the injection of priors regarding the desired motion captured from large-scale pretraining, and encourages the agent to achieve different poses corresponding to the same text input. As the "most canonical" pose that aligns with the text caption as judged by the pretrained diffusion model varies across source noise samples, the agent is naturally disincentivized to remain motionless in one particular configuration, and instead is motivated to keep moving in a fashion consistent with the provided text description.

Attempting to verify the hypothesis by running CLIP on sequences of Walker2d-v4 trajectories yields the observation that a particular pose is consistently selected as highly-aligned with the text prompt, which is similar to the pose achieved by the final video produced by a trained CLIPSimPol model. On the other hand, as TADPol leverages a large-scale generative model, it potentially has the ability to generalize to a wider variety of visual settings and a deeper understanding of how different poses relate to a consistent text prompt. Then, rather than finding a deterministic canonical pose, it encourages the agent to consistently achieve multiple poses that align well with the text prompt. We believe that this fundamental difference causes TADPol to be more suitable to convert text inputs to temporally extended behavior than CLIPSimPol. A visualization of this finding is included in Figure 3.

## A.2  Variance Penalization and Reward Augmentation

As mentioned in Section 4.3, there are fundamental conceptual limitations in generating a reward signal from purely a static text-to-image model. Despite being less likely to occur than in CLIPSim-Pol, it is still possible that the agent discovers a single frame that well-aligns with the text caption so much so that it is no longer incentivized to move. Such a case would then produce a collapsed video, where each frame is identical to the last. Variance penalization is one potential way to prevent this from occurring, and keep the agent in perpetual motion. Variance penalization is an additional term that is subtracted from the reward signal at each time step, and is calculated from past, achieved alignment scores. With variance penalization, the new reward that is provided to the agent via TAD-Pol is:

$$r_t = \exp\left(-\kappa(t_{noise}) \left\| \boldsymbol{\epsilon}_0 - \hat{\boldsymbol{\epsilon}}_\phi(\tilde{\boldsymbol{o}}_{t+1}; t_{noise}, y) \right\|_2^2\right) - c_1 \exp\left(-c_2 Var(r_{\leq t})\right)$$

Here, $c_1$ and $c_2$ are tunable constants to control the effect and scale of the variance penalization term. As the variance increases, the penalization becomes smaller; however, when the agent becomes lethargic, the term kicks in and encourages movement. Variance penalization is task and environment agnostic, as it simply operates off of computed alignment scores, and can be applied to arbitrary TADPol implementations. Beyond discouraging an agent from remaining in a particular state or configuration for too long, it has the additional bonus of sifting out unaligned text prompts. Text-prompts that are unaligned with the environment or agent of interest will naturally produce low mean and low variance alignment scores, as judged by the pretrained diffusion model, regardless of how the agent interacts with the environment. Adding a variance penalization term will encourage agents to learn to terminate the episode as soon as possible, when provided an unaligned prompt. This allows for the learning of behaviors only when the text prompt is well-aligned with the capabilities and visual properties of the environment. We provide video demonstrations of TADPols trained

with a Variance Penalization term, and demonstrate that they learn coherent, text-aligned motions for informative text prompts, but learn to terminate immediately for unaligned text inputs.

Another fundamental limitation of using a text-to-image model springs from the fact that the supervision comes entirely from a rendered image and a text prompt. However, it is often desirable to learn a policy that considers non-visual factors as well, such as minimizing control costs or the energy expended by the agent as it interacts. Such information may not be naturally extractable from the rendered image alone, even if such desiderata are specified in the form of natural language. To address this, TADPol can easily be augmented with other available reward signals that consider factors beyond the image domain, such as through simple arithmetic combination. Used in such a way, the role of TADPol becomes a way to inject natural language priors to guide the behavior of the agent from a visual perspective. In conjunction with other alternative signals, such as speed or the energy expended, the resulting agent can then be optimized to accomplish its goal while visually aligned with a provided natural language text prompt. We anticipate extensions of TADPol to be exciting future work to explore.

## B  REBUTTAL EXPERIMENTS

We perform experiments to demonstrate the ability of TADPol to learn a variety of novel behaviors directly from text conditioning. In particular, we apply TADPol to both goal-achieving tasks, such as striking a particular pose, as well as continuous locomotion tasks. It should be noted that achieving continuous locomotion from text conditioning is a difficult endeavour that has not been thoroughly explored in prior work to this author's knowledge; whereas in goal-achievement a desired final goal state can be created or selected, and the task becomes a form of inversion to achieve it, in continuous locomotion tasks such as "walking", "rolling", or "dancing", there is no canonical pose or frame that, if achieved, represents a completion of the task. Instead, such activities have to continuously cycle through multiple motions (swinging legs in a coherent sequential motion, for walking) to accurately constitute satisfying the desired behavior.

In this work we demonstrate that TADPol is a naturally more promising approach to tackle continuous locomotion, on top of being able to handle goal-achieving objectives. We use the advanced, complex Dog environment from the DeepMind Control Suite to showcase TADPol's abilities. The Dog agent has a large degree of freedom for motion, with no task-specific priors (such as termination conditions), providing a flexible and expressive testbed to convincingly show how TADPol can flexibly learn novel behaviors specified by natural language. For our experiments, we utilize TADPol as the supervisory signal to train a TD-MPC agent Hansen & Wang (2021), the first documented model capable of solving ground truth Dog tasks. We demonstrate that through TADPol, novel behaviors beyond those specified by ground truth reward signals can be learned through text conditioning.

To better align the environment with the priors of large models pretrained on internet-scale and natural data, we modify the terrain of our MuJoCo rendered environments to have green grass. This particular choice was selected randomly, and different terrain textures and aesthetics were not ablated over. The Dog agent itself, in terms of its underlying state as well as its visual appearance, was completely untouched. Furthermore, all TADPol design decisions, such as noise level, were kept consistent with what was described in Section 4, unless otherwise noted.

In these experiments, we introduce a more comprehensive set of baselines. We compare TADPol against other text-to-reward approaches, including CLIP-SimPol Rocamonde et al. (2023), LIV Ma et al. (2023), and Text2Reward Xie et al. (2023). CLIP-SimPol is concurrent work , having been submitted to the same conference, and is already compared against in the prior experiments of this work. Furthermore, the authors of the concurrent work only evaluate CLIP-SimPol on goal-achieving tasks, whereas we go beyond it to also evaluate continuous locomotion capabilities. We also benchmark TADPol against a checkpoint from LIV, which was mentioned in the related work as a technique to generate dense rewards from text specification. We also evaluate against Text2Reward, another work submitted to this conference, which generates a reward function for a desired text-specified behavior using a pretrained LLM. In these experiments, we prompt ChatGPT with a similar template as used in Text2Reward for Hopper, with the details of the particular Dog agent specified instead of the Hopper agent. It is of note that Text2Reward does not have access to visual information, as it is interfacing with a pure language model, but has access to real-time state signals such as speed and direction (which the other approaches, including TADPol, do not).

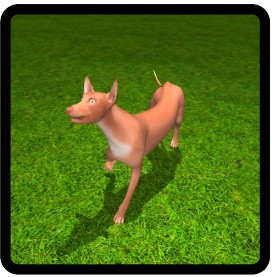
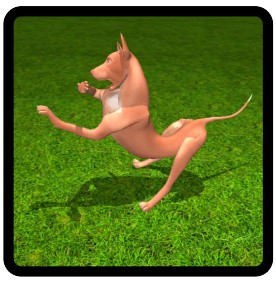
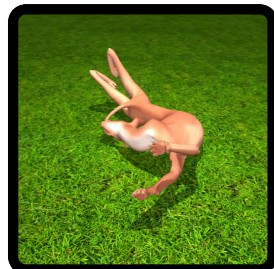

"a dog standing"        "a dog standing        "a dog chasing its tail"
on hind legs"

Figure 4: We show that TADPol can successfully learn goal-achieving policies. Furthermore, TAD-Pol demonstrates sensitivity to subtle variations to the input prompt, learning different behaviors between "a dog standing" and "a dog standing on hind legs". We visualize selected frames from the resulting policies learned by TADPol; full videos are available on the associated website.

## B.1 GOAL-ACHIEVEMENT

We begin by learning goal-reaching tasks via TADPol, and benchmark it against other text-to-reward approaches. We first task the Dog agent to stand upright, with the prompt "a dog standing". Although there is no ground-truth reward function that purely promotes natural standing behavior for the Dog with which to use as an evaluation signal, a few insights can still be derived from the resulting policies learned. Firstly, we observe from the provided videos of the final trained policies that TADPol is indeed able to learn how to maintain a dog in a natural upright position on all four legs. On the other hand, CLIP-SimPol learns to achieve and maintain a stable position for the dog, but in a rather contorted pose. This may be because it interprets the word "standing" as erecting oneself vertically, as humans do, but fails to generalize to how dogs in particular naturally "stand". A similar observations hold for the result of Text2Reward, which learns to balance a dog stably on three legs, whereas LIV completely fails to learn a standing policy whatsoever. Qualitatively, it appears that the pose maintained by TADPol is the most aligned with the intention behind the "a dog standing" prompt. We therefore believe that the visual priors within a pretrained diffusion model, leveraged in conjunction with their strong alignment capabilities with natural language, are not only capable of but indeed a preferred choice for learning natural-behaving policies conditioned on text descriptions.

We further investigate whether or not TADPol is sensitive to subtle variations of the input prompt. We therefore change the conditioning phrase from "a dog standing" to "a dog standing on its hind legs", and train an agent as before. In Figure 4, we visually verify that the resulting dog policy indeed learns to balance on its hind legs, and even tries to recover such a position if it loses balance. This is a novel behavior for which there is no ground-truth defined reward function; yet, TADPol enables us to learn a policy to behave in such a way conditioned on natural language alone. Furthermore, as another novel goal-achieving behavior, we showcase results for the prompt of "a dog chasing its tail", and demonstrate that the resulting policy indeed learns a dog that seeks to bite its tail. We take these findings as strong signals for two insights: that TADPol can indeed unlock the learning of flexible and natural-looking behaviors from text, and that TADPol is able to respect fine-grained details and subtleties of the input prompts. We encourage the readers to evaluate the quality of the resulting videos for themselves, which are available on the updated website (https://sites.google.com/view/tadpol-iclr24/).

## B.2 CONTINUOUS LOCOMOTION

We further explore the ability of TADPol to learn continuous locomotion behaviors conditioned on natural language specifications. As previously mentioned, such tasks are difficult to learn purely from external description, as there is no canonical pose or goal frame that if reached, would denote successful achievement of the task. For example, there is no specific pose for "running" that when reached, would successfully represent completion of the "running" behavior; instead, agents must perform a coherent continuous sequence of poses over time (such as alternating stepping with the legs and swinging the arms) to convincingly demonstrate accomplishment of the task. This is naturally challenging for approaches that essentially statically select a canonical goal-frame to achieve,

Table 3: A comparison between the resulting policies trained via CLIPSimPol, LIV, Text2Reward, and TADPol for the "a dog walking" prompt conditioning, as measured by a ground truth reward signal for "running". We discover that compared to other approaches, diffusion-based models provide a superior reward signal to supervise the learning of motion behavior in policies.

| Method | Ground-Truth Reward (Running) |
|---|---|
| CLIPSimPol | 53.19 |
| LIV | 16.86 |
| Text2Reward | 63.15 |
| TADPol | **84.66** |

such as CLIP or LIV, and we propose TADPol as a promising direction forward. Indeed, the static nature of the goal-frame selection through CLIP-like approaches, and detailed analysis into why they struggle to supervise the learning of a policy for continuous locomotion, is expounded upon in Section 4.2 and Appendix A.1.

We begin by evaluating the policies learned from conditioning on the prompt of "a dog walking", across various choices for the text-to-reward signal. As a ground-truth reward functions for "walking" and "running" are defined in the Dog environment, this choice of prompt enables us to directly and quantitatively compare results across different approaches in terms of how well they can recreate the ground-truth specified behavior; in our experimental metrics we evaluate using the "running" ground-truth reward signal since it can evaluate a wider range of speeds due to its higher maximum threshold. We first notice that the CLIP-SimPol approach collapses; it learns to strike a particular pose and maintains it. Remaining motionless but upright, the ground truth reward it achieves is only 53.19. On the other hand, LIV fails to learn a walking motion completely and achieves a reward of 16.86. Text2Reward, which does not have access to any visual information, but does have access to ground-truth state information for the dog including speed, direction, and joint positions, achieves a reward of 63.15. Ultimately, TADPol achieves the highest reward amongst all choices with 84.66, while also distinguishing itself as the most natural-looking policy in qualitative terms. The resulting Dog moves its legs in plausible step-taking motions while keeping itself upright. Although the Dog does not make substantial progress in terms of ground distance, it clearly learns to produce movement over static goal-achievement in contrast to CLIP-based approaches, and the movements it learns appear to all relate to the provided text prompt. We take this as a positive signal that TADPol is not only able to learn goal-achievement policies, but is also favorable for learning policies to perform text-conditioned continuous locomotion.

### B.3 TADPOL DESIGN DECISIONS

Interesting design decisions pertaining to TADPol chiefly revolve around the source noise, which is itself a critical component of diffusion models. How often the source noise is re-sampled is a natural way to affect the behaviors learned by TADPol. Intuitively, for a fixed noise perturbation level and text conditioning, it is the source noise vector that controls what predicted clean image the pretrained, frozen diffusion model produces. If the source noise vector is sampled once and kept global, then the expected behavior would be similar to CLIP-SimPol, as there is one canonical frame that if achieved by the agent, should yield the least prediction error when denoised through the pretrained frozen diffusion model. The agent then seeks to find and maintain the frame that is essentially determined by that particular, static source noise vector. On the other hand, routinely re-sampling the source noise is a mechanism through which the goal-state updates, thereby promoting motion, where all goal-states share high-alignment with the text conditioning as they are selected implicitly through the denoising procedure of the pretrained diffusion model. We experimentally find that re-sampling the source noise every episode is a happy balance; re-sampling for each frame can cause "schizophrenia" in the model, as the high-variance and instability of the reward function can lead to unstable policy behaviors, and having a global source noise would behave similarly to CLIP-based approaches. We indeed provide visual demonstrations in the updated website that a global source noise results in a stable Dog agent, whereas the per-frame re-sampling approach results in a more unstable Dog that is even prone to falling down.

Another interesting design choice is how much noise level to apply to the rendered clean image. We discover that when using too high a noise level, such as a timestep of 750, the Dog agent fails to learn coherent behaviors. Visualizing the noising procedure, we find that at such a high level all original structure of the input image is corrupted to the point of unrecognizability; denoising from such a high noise level simply amounts to conjuring text-conditioned structure from almost no original signal. This is not useful for calculating the alignment between the rendered image and a provided text prompt, which necessarily requires some degree of original structure intact, and helps to preliminarily explaining the subsequent difficulty in learning a coherent policy. We discover that a "sweet spot" for the noise level exists around 450 that sufficiently balances noise corruption while avoiding overcorruption, and use the level of 450 for all our experiments.

## B.4    AUXILIARY REWARDS

As mentioned in Appendix A.2, a natural limitation of using individual frames to generate dense rewards is that it is fundamentally unable to determine temporally-extended concepts such as speed or traveling direction. Furthermore, it is often desirable to integrate non-visual factors into a reward function as well, such as minimizing control costs. In this work we demonstrate preliminary experiments into the performance of TADPol combined with auxiliary rewards. In particular, we add a simple, unscaled horizontal velocity reward, bounded between 0.2 and 1.0, to TADPol at each timestep. The velocity reward encourages the dog agent to move quickly along any arbitrary direction. We showcase this hybrid reward for a prompt of "a dog walking" and demonstrate that the dog indeed learns to make horizontal progress on top of visually appearing to walk. Furthermore, we learn a prompt of "a dog rolling on the ground" with the simple velocity reward, and show that the learned behavior of the dog agent meaningful changes. The resulting policy learns to move along the ground, but with the dog's head and body low and close to the ground, simulating rolling on the ground. As a baseline, we visualize a Dog agent that has been trained on the speed reward alone, which does not naturally manifest itself in an upright running motion, and show that TADPol meaningfully adapts the behavior of the Dog agent to respect text conditioning in a visual way. We therefore demonstrate that TADPol can be cheaply and flexibly extended with auxiliary rewards to learn novel behaviors that simultaneously respect multiple desiderata, such as speed and natural language conditioning.

