# OpenReview forum: "Text-Aware Diffusion Policies"
_ICLR.cc/2024/Conference — Submitted to ICLR 2024_

### Official Review · Reviewer_JZP2 · 2023-10-13

**Soundness:** 4 excellent
**Presentation:** 3 good
**Contribution:** 2 fair
**Rating:** 6
**Confidence:** 2

**Summary:**

This paper presents a policy learning method that uses text-visual alignment as a reward signal. Diffusion models are used to compute the matching scores between text and images. The authors then conduct experiments on a variety of locomotion experiments, and the results show that the proposed method can learn to perform following the text while performing the motion as autonomous agents.

---

Post rebuttal: the authors provided extensively more experiments, which not look very good but it's reasonable. As a result, I upgrade my score from five to six.

**Strengths:**

1. The idea of adopting diffusion models as a reward model is novel in my feeling. I think this idea may have other applications and this is very insightful.
2. I love the provided video demos. They demonstrate that the proposed method can work in several environments.
3. The proposed approach outperforms a CLIP-based method, which is promising and it indicates that diffusion models may be a better measurement model.

**Weaknesses:**

I have a few concerns in my mind. However, I'll re-rate this paper after the rebuttal and after seeing other reviews.
# The soundness of this work.

1. The game environments are not natural images but the diffusion models are trained on natural images. How will this work in a game environment? I doubt the effectiveness of diffusion models in such a scenario. Can the authors show some generated images in this locomotion environment? Otherwise, I doubt the effectiveness of using diffusion models here.

2. If the video is not used, why doesn't the model get stuck into some best-matching frames? Since policy learning often involves some non-trivial procedures, it is not convincing if the authors only use text-to-image techniques but not video encoding methods. The matching should be conducted in the temporal space.

3. I see that the provided video demos do not have a very complex procedure. Most videos are nearly static and do not move much. This problem may be due to the use of images in the reward calculation.

# Comparisons and experiments.

1. Why don't the authors compare their methods in other language-guided tasks such as language-guided navigation?

2. The results shown in Table 1-2 are not very surprising. The authors are recommended to try harder problems. Also, comparing it to pure RL methods is a must.

**Questions:**

1. Why do the authors construct a large model for shaping rewards? This requires a good explanation.

2. How reliable the noise is to calculate the reward? Sometimes the noise may not mean much thing.

---

> ### Author Response · Authors · 2023-11-21
> **Response to reviewer JZP2**
>
> We thank Reviewer JZP2 for their thorough review.  We seek to address their listed considerations below:
>
> - On soundness: the reviewer raises the question of how diffusion models can be applied to frames produced by game environments, if the diffusion models are trained on natural images.  Indeed, the diffusion models are trained on natural images; however, they are also trained on cartoon images, paintings, movie posters, and more.  We believe that the large-scale, pretrained diffusion model is able to encode the essence of the environment.
>
> - On avoiding getting stuck: we provide analysis into why leveraging a text-to-image generative model as a reward signal does not cause the resulting policy to get stuck in achieving some best-matching frames in Appendix A.1.  On the other hand, this is a common and natural pitfall when using alignment models such as CLIP as the supervisory reward signal to train a policy, as we both quantitatively and qualitatively demonstrate throughout the paper (in particular Walker, as well as the new Dog experiments).  We believe that the stochasticity stemming from the source noise causes the pretrained, frozen diffusion model to constantly change its current belief of what the "best-matching frame" is.  However, this does not result in a wildly unstable reward signal; as the diffusion model has learned a conditional *distribution*, these changing "best-matching frame" beliefs will always have high alignment with the provided text prompt.  Ultimately, for continuous locomotion prompts, where there is a large set of aligned images (e.g. for "walking" there are many valid poses of different configurations of swinging arms and legs), we find that TADPol learns to perform coherent, text-aligned motion.  On the other hand, as CLIP is deterministic, there is inherently a singular frame achievable by the policy that will have the highest alignment score with the text prompt.  Then, a policy optimized using a pretrained CLIP checkpoint will learn to seek and get stuck maintain that best-matching frame.
>
> - On using video models: we would like to mention that we do compare against video methods: we perform experimentation of our approach against Vid-TADPol, which uses a text-to-video diffusion model in place of a text-to-image one for TADPol.  However, we surprisingly find that Vid-TADPol achieves comparable performance with TADPol in our experiments.  One potential explanation might be that learning to walk in the OpenAI Gym environment is relatively insensitive to the priors encoded in the text-to-video model beyond those found in a text-to-image model.  Another potential explanation is that Vid-TADPol is limited by the quality of currently existing text-to-video solutions.  Addressing this finding and exploring how natural videos can be used to supervise the learning of novel text-conditioned policies is exciting future work.
>
> - On the complexity of video demonstrations: we provide additional results demonstrating TADPol on the complex Dog environment from the DeepMind Control Suite.  In these experiments, we show that TADPol is able to learn complex procedures (such as having the Dog stand on its hind legs or chase its tail).  Furthermore, we showcase in our updated experiments how TADPol can learn motion policies.
>
> - On comparisons to language-guided tasks: in this work we focus our efforts on learning text-conditioned continuous locomotion capabilities, and we show evidence suggesting TADPol is a promising approach towards this objective.  Text-conditioned continuous locomotion is difficult compared to goal-achieving tasks (including language-guided navigation) in that there is no canonical pose or goal frame that, if reached, would denote successful achievement of the task.  For example, there is no specific pose for “running” that when reached, would successfully represent completion of the “running” behavior; instead, agents must perform a coherent continuous sequence of poses over time (such as alternating stepping with the legs and swinging the arms) to convincingly demonstrate accomplishment of the task.  To our knowledge, there are no prior works that seek to tackle this objective.  Furthermore, the environments of many language-guided tasks are egocentric in nature; we leave it as interesting future work to explore whether or not pretrained diffusion models naturally understand how intermediate egocentric frames should be aligned with a desired text instruction.

---

> > ### Author Response · Authors · 2023-11-21
> > **Response to reviewer JZP2 (cont.)**
> >
> > - Comparing against pure RL methods: our investigation focuses on learning flexible, novel behaviors specified by text input, beyond the behaviors defined by ground-truth reward functions.  Comparing against RL methods for such behaviors naturally means defining an accurate ground-truth reward function conditioned from text, and then training a vanilla RL policy on it.  In our new experiments, we perform this exact setup, where the ground-truth reward function is generated from a textual description of the behavior through a LLM, as proposed by Text2Reward.  We find that Text2Reward underperforms compared to TADPol in both quantitative and qualitative comparisons, and across both goal-achieving and continuous locomotion objectives.  A detailed comparison, including numerical, is provided in the updated manuscript in Appendix B.1 and B.2, and visualizations are provided on the updated website.
> >
> > - On utilizing a large model for rewards: to avoid potential confusion, the $k(t)$ term is not a neural network; it is a scaling parameter that can optionally change with respect to the level of noise $t$ used.  For our experiments we use $k(t) = \frac{1}{2} (1 - \prod_{i \leq t} \alpha_i )^2$; however, we note that in our experiments $k(t)$ can essentially be treated as a constant, since we only evaluate with a constant t (where $t=450$ in our experiments).  However, we include a $k(t)$ term in the TADPol equation to keep it general, allowing it to flexibly change with respect to $t$; in experiments where $t$ is resampled during training, this term provides the option to flexibly adjust the reward computation accordingly.  If the reviewer was wondering why a large model, namely a large-scale pretrained text-to-image diffusion model is useful to generate rewards, we believe that it is a promising way to scale up reinforcement learning models and unlock the learning of policies that align with priors and properties discovered from large-scale data.  Reinforcement learning development has long been isolated from the scaling benefits of data and architecture observed in other domains, such as vision, language, and their intersection.  By leveraging such models appropriately as a reward signal to supervise the learning of policies, priors from large-scale pretraining (such as what visually occurs naturally) and properties (such as strong multimodal or crossmodal alignment) can be transferred to expand the limits of what can be achieved through reinforcement learning.  For example, in this work, we demonstrate how such large-scale pretrained models can be leveraged to unlock the learning of policies that behave in alignment with text conditioning.
> >
> > - On the importance of noise: the noise is actually quite critical to the behavior of TADPol.  Conceptually, for a given source noise, there is a canonical frame that would have least reconstruction error as judged by the pretrained frozen diffusion model.  For different source noise samples, the canonical frames change; however, they are still judged with respect to the text conditioning through the diffusion model.  It is therefore an interesting ablation study into how often to resample the source noise for best performance results.  In the updated experiments, as presented in Appendix B.3, we demonstrate that having a global source noise essentially learns CLIP-like behavior, since the canonical frame that minimizes the predicted error is fixed for the entire training.  We also demonstrate that resampling the source noise for each frame can lead to instability in the reward function and therefore the learned policy.  Re-sampling the noise vector for each episode shows stable behavior.  We also experiment with varying strengths of noise corruption to the clean source image, and show that applying too high a noise level results in degraded performance for the resulting policy.

---

> > > ### Comment · Reviewer_JZP2 · 2023-11-22
> > > **Thanks for your response.**
> > >
> > > The provided experiments on dogs do not look very good. Those dogs move very awkwardly, which differs from realistic dogs very much. Also, the stochasticity cannot fully account for the good motion, because the direction of stochasticity cannot be controlled well.

---

> > > > ### Author Response · Authors · 2023-11-22
> > > > **Thank you for the updated comments.**
> > > >
> > > > We thank Reviewer JPZ2 for the prompt reply.  As for the quality of the Dog movement, we believe this speaks to the challenging nature of the locomotion task itself; with a large-dimensional (38) continuous action space, even learning to run with a **ground truth reward** associated with the environment has been a long-difficult task (with many established approaches such as SAC failing - [example](https://nicklashansen.github.io/td-mpc/videos/dog-run-sac.mp4)). Moreover, our TADPol with the “a dog walking” prompt ([hybrid mode](https://drive.google.com/file/d/1MBx1JcKYiQuscjNO6Bt2d0Z1UMAj9bCo/view?usp=drive_link)), achieves a similar level of visual quality as the TD-MPC model with the environment's **ground-truth** reward for ["walking"](https://nicklashansen.github.io/td-mpc/videos/dog-walk.mp4) or [“running”](https://nicklashansen.github.io/td-mpc/videos/dog-run.mp4), without relying on the task-specific, manually designed nature of the provided reward functions. In addition, for other tasks such as goal-achievement, we can visually verify that the Dog indeed learns to achieve the specifications from the text (such as standing on its hind legs, or chasing its tail) purely from text conditioning alone without the ad-hoc design of ground-truth reward functions.  Put in the context of the difficulty of the Dog experiments, we believe this successfully showcases the ability of TADPol to provide expressive, meaningful rewards from natural language descriptions.
> > > >
> > > > We agree that the direction of stochasticity cannot be controlled; however, we experimentally and conceptually show that our approach outperforms other techniques that have no stochasticity whatsoever (such as CLIP-based approaches).  Using a pretrained conditional generative model provides the guarantee that the policy will always strive to achieve poses aligned with the text-prompt for arbitrary source noises; then, the stochastic nature of the formulation helps to produce motion and avoids getting stuck in a fixed goal-state.  TADPol is thus a promising direction to learn text-aligned motion, as even if the exact stochasticity cannot be controlled temporally, the resulting motions that TADPol learns to produce are all aligned with the text prompt in some way.  We demonstrate this both qualitatively and quantitatively; all other baseline results (such as CLIP-SimPol and LIV) refrain from moving, whereas TADPol shows higher text-aligned movement.
> > > >
> > > > We can further highlight evidence of TADPol's ability to learn text-aligned motion through the hybrid experiments (best visualized in the updated website).  We show that when adding a simple task-agnostic speed term, the resulting policy indeed learns to make the dog walk by alternating its legs in a natural motion ([visualization](https://drive.google.com/file/d/1MBx1JcKYiQuscjNO6Bt2d0Z1UMAj9bCo/view?usp=drive_link)).  We also show that this speed term is not the main magic behind this natural-looking walking motion; we demonstrate that when training solely on the speed term, the dog does not take natural-looking steps ([visualization](https://drive.google.com/file/d/1vYUwDU2Wl7Jv9oq75w_BNMbmb6v9CHtI/view?usp=drive_link)), nor does it take natural-looking steps when provided with a text prompt of "a dog rolling on the ground" with the speed term ([visualization](https://drive.google.com/file/d/1g9lhxwflNVr0OKYuRTw1Hs66FWWdLX4m/view?usp=drive_link)).  Therefore, we believe the learned walking motion indeed stems from the text-conditioned prompt of "a dog walking", highlighting TADPol's ability to convert natural language into expressive and flexible reward signals.

---

> > > > > ### Comment · Reviewer_JZP2 · 2023-11-22
> > > > > **Thanks for your response again.**
> > > > >
> > > > > 1. Try to pick some good demos. Although some envs may not be very satisfying, the proposed approach should do well in some other environments.
> > > > >
> > > > > 2. The authors should compare to others that adopt diffusion as policy, not the methods running without diffusions. These are good baselines (mentioned by another reviewer): https://arxiv.org/abs/2310.12921 https://arxiv.org/abs/2203.12601 https://arxiv.org/abs/2210.00030. Also, I suggest the authors compare to the concurrent work https://openreview.net/forum?id=N0I2RtD8je. I read your comments that you said that the settings are somewhat different. Yes, you experimented in different setups, but their setups were making more sense (e.g. with imitation learning and diffusions for exploration). So comparisons were still needed. It's not only an issue of choice of baselines, I think moving to their setups is also benefit for current manuscript.

---

> > > > > > ### Author Response · Authors · 2023-11-22
> > > > > > **Thank you for the consistent responses**
> > > > > >
> > > > > > We thank Reviewer JZP2 for their consistent responsiveness and helpful suggestions on our work.  We are grateful for their clear intention to improve our submission.
> > > > > >
> > > > > > ### **On Demonstrations:**
> > > > > >
> > > > > > We selected the Dog environment for its known difficulty in the RL space, in response to a consistent interest from the reviewers in TADPol's capabilities on more advanced environments beyond the OpenAI Gym demonstrations.  We demonstrate it can achieve comparable quality with models trained with a ground-truth reward signal, alongside new-found text-conditioned behaviors such as novel goal-achievement.  Across the environments selected, we also demonstrate consistent outperformance against other text-conditioned methods such as CLIP-SimPol.
> > > > > >
> > > > > > Whereas the reviewer may not be satisfied with one particular environment, we agree with the reviewer's belief that TADPol should do well in other environments (and have shown this to be the case via Hopper, Walker, etc.).  Indeed, these additional experiments further confirm the versatility of TADPol's ability to learn text-aligned policies across multiple environments, with varying visual domains, and for complex robots with high-dimensional states.

---

> > > > > > ### Author Response · Authors · 2023-11-22
> > > > > > **Thank you for the consistent responses (cont.)**
> > > > > >
> > > > > > ### **On Baseline Comparisons**:
> > > > > > Fundamentally, TADPol proposes a new way to generate *rewards* that are text-conditioned, and is completely agnostic to the choice of policy implementations beneath it.  Indeed, TADPol can be applied **on top** of methods that use diffusion as a policy, rather than in direct comparison against them.  We therefore focus our experimental comparisons across other methods that provide dense rewards from natural language descriptions, as all of these models can be applied regardless of the choice of policy network below them.
> > > > > >
> > > > > > In this context, we agree with this reviewer on the relevancy of the listed baselines (and note that the concurrent work on OpenReview is the exact same work as the first arXiv baseline suggested).  We would like to highlight that we **already** provide experimental comparisons with two of them.  The first baseline work/OpenReview submission is implemented in our work as the CLIP-SimPol baseline, and was compared against for all experiments (including Hopper, Walker, etc.).  The details of the CLIP-SimPol method that we independently proposed are the exact same as what was provided in the concurrent work, where the reward is simply calculated using a pretrained CLIP checkpoint.  In our experiments, we also use the same pretrained CLIP checkpoint as the one used in the concurrent work for fair comparison.  We note that this concurrent work was not publicly made available until after the original submission deadline, but in this updated draft of the manuscript we have since added references to them for our CLIP-SimPol baseline.  We also quantitatively and qualitatively improve over this baseline across our experiments, and provide conceptual analysis and justification into why this is the case by exposing the inherent limitations of CLIP-SimPol (Section 4.2, and Appendix A.1).
> > > > > >
> > > > > > The third baseline listed by the reviewer, VIP, is not directly comparable to our work since it does not learn text-conditioned behavior.  In VIP, tasks are specified through goal images, whereas in our work we are interested in learning tasks specified by natural language.  However, we highlight LIV, a follow-up work to VIP, as a way to generate dense rewards from text specification. In our additional rebuttal experiments on the Dog environment, we do include comparisons against LIV, and both qualitatively and quantitatively show that it underperforms on both goal-achieving and continuous locomotion tasks compared to TADPol.  We therefore do provide indirect comparison against this baseline, by comparing against its follow-up method adapted to text-conditioned reward generation.
> > > > > >
> > > > > > We note that the second proposed baseline listed by the reviewer, R3M, is limited to imitation learning, and was not at all applied to reinforcement learning settings.  In this work, we focus our efforts on learning text-conditioned policies for reinforcement learning.  In imitation learning, the behavior specification is provided in the form of an explicit trajectory to follow.  However, this assumes the existence of an expert trajectory demonstration already, which may be expensive to generate, and must be gathered for each new environment or robot.  In contrast, we seek to perform text-conditioned policy learning, where the behavior specification is provided in the form of a natural language description.  Language is a natural and flexible interface with which to express desired behavior across arbitrary environments or robots, and does not require an expert to first perform the behavior for imitation.  We therefore focus our comparison of TADPol against other methods that learn reinforcement learning policies from text-conditioning such as CLIP-SimPol, LIV, and Text2Reward, and show that TADPol outperforms them in both goal-achieving and continuous locomotion tasks.
> > > > > >
> > > > > > We are curious to which work the reviewer is referring to with respect to the comment on diffusions for exploration, as none of the listed papers seem to use or reference diffusion in their manuscripts.  We point out that none of the prior baselines listed compare against specific policy or exploration choices, as all of them (in addition to TADPol) are capable of being applied on top of *arbitrary* exploration techniques or policies.

---

> > > > > > > ### Comment · Reviewer_JZP2 · 2023-11-23
> > > > > > > **Thanks for the additional feedback.**
> > > > > > >
> > > > > > > 1. Authors say, "Dog is a known environment in RL space". It's better to provide some reference. I haven't seen any papers in the RL community using this environment. It's better to use some robotic environments instead.
> > > > > > >
> > > > > > > 2. About baselines: CLIP-SimPol is not that important in my eyes because it's not a standard policy learning method. The crucial point is to compare against three lines of work: RL, imitation learning, and unsupervised learning. Currently, CLIP-SimPol can fall into the third category, but results of standard RL and imitation learning are missing. It's interesting to see whether the proposed approach can benefit RL and imitation learning methods. Authors criticize that  "this assumes the existence of an expert trajectory demonstration already." I don't agree with such a criticism because imitation learning is a vast domain, and it can often address problems that RL cannot address. Also, learning from a big data is a good general direction and I think imitation learning will have a good future.

---

> ### Author Response · Authors · 2023-11-23
> **Thank you for the additional comments.**
>
> We thank Reviewer JZP2 for the prompt and informative replies, particularly this late in the discussion period, and seek to address their concerns.
>
> **On the difficulty of the Dog environment:** we provide two references, [1] and [2], that testify to the difficulty and complexity of the Dog environment, particularly due to its large 38-dimensional action space and its complex transition dynamics.  Concurrent work (https://openreview.net/forum?id=Oxh5CstDJU) demonstrate that SAC and DreamerV3, powerful model-free and model-based approaches established in the field, are unable to learn any of the Dog tasks despite training on the ground-truth reward ([Figure 5](https://openreview.net/pdf?id=Oxh5CstDJU)).  We selected the Dog environment to demonstrate TADPol's capabilities because beyond being complex, it naturally enables us to flexibly demonstrate a variety of goal-conditioned *and* continuous locomotion behaviors.  We agree that robotic environments are similarly worthwhile and interesting to apply TADPol to, and leave it as interesting future work.
>
> **On comparisons with RL:** We agree with the reviewer that our method is RL that uses external supervision, and we do indeed compare against other externally-supervised RL techniques (the authors believe that "unsupervised RL" as a term might more accurately describe curiosity-based or intrinsic motivation-based RL, where the agent generates its own reward signals).  We also agree with the reviewer that comparison with standard/ground-truth RL is valuable; and indeed we do already provide such numerical reports in Column 2 of Table 1 of the paper.  For further reference, in the updated experiments on the Dog, the ground-truth TD-MPC model achieves about 150 reward, whereas our hybrid TADPol Dog agent achieves a comparable reward of 138; the rest of the ground-truth rewards achieved by externally-supervised approaches are listed in Table 3.  Quantitatively, we show that TADPol is able to achieve high ground-truth reward despite using an external text-conditioned reward as a supervisory signal; qualitatively, we show that it can indeed recreate the behaviors described by the original ground truth reward function.  We thus believe TADPol is comparable to ground-truth RL techniques.  We also note that ground-truth RL techniques require detailed ad-hoc design of the reward function - in the absence of such ground truth reward functions, such as making a Dog stand on its hind legs or chase its tail, we demonstrate TADPol's superior performance against other externally-supervised techniques such as LIV, CLIP-SimPol, and Text2Reward.
>
> **On comparisons with IL:** We would like to clarify that we wholly support the development of imitation learning as a field; it is certainly an exciting, interesting, and valuable direction of study.  However, we are unable to apply imitation learning techniques for the Dog environment, as there are no ground truth demonstrations for the Dog tasks (such as for chasing its tail or standing on its hind legs).  In these cases, without an appropriate dataset of expert demonstrations, imitation learning cannot be performed and compared against.
>
> We believe that when expert demonstration datasets are available, prescribing imitation learning as a solution is appropriate and desirable.  We are also interested in further developments in this field, and agree that it has a bright future.  On the other hand, RL approaches are complementary to this direction, as they are able to be applied when demonstrations are scarce or nonexistent; in such cases, we demonstrate that TADPol can be applied to learn novel behaviors conditioned flexibly on natural language.
>
> [1] Yi Zhao et al. Simplified Temporal Consistency Reinforcement Learning. ICML, 2023.
>
> [2] Nicklas Hansen et al. Temporal difference learning for model predictive control. ICML, 2022.

---

> ### Comment · Reviewer_JZP2 · 2023-11-23
> **I'll update my score.**
>
> Thanks for your response. I also understand the effort of the authors to publish this paper in time. Some of my concerns are addressed, and the response reads reasonable. So I'll upgrade my score by one.

---

> > ### Author Response · Authors · 2023-11-23
> > **Thank you for your consideration.**
> >
> > We thank Reviewer JPZ2 for all their time, energy, and clear commitment to improve our submission throughout this discussion period.  We are happy to hear that we were able to address some of their concerns, and appreciate the improvement in score.

---

### Official Review · Reviewer_gAGE · 2023-10-31

**Soundness:** 3 good
**Presentation:** 3 good
**Contribution:** 2 fair
**Rating:** 3
**Confidence:** 3

**Summary:**

In this paper, the authors propose Text-Aware Diffusion Policies (TADPols), which attempt to learn the text-aware policy through reward function with the help of the generative prior contained in pre-trained text-image models. In particular, the reward function measures the alignment of provided instructions and images rendered of agent actions. The authors compared the proposed framework with baselines that use a reward function defined by CLIP similarity or using a text-to-video diffusion model.

**Strengths:**

- Overall, the paper is well-organized and easy to follow.

- The idea is intuitive and straightforward. The authors defined an explicit and simple reward function to optimize the policy.

**Weaknesses:**

- The authors need to compare with other works such as LangLfP, Text-Conditioned Decision Transformer, or Hiveformer.

- As mentioned in section 4, the choice of noise step might be sensitive to the performance of the proposed method. It would be helpful if the authors could provide some experiments on the choice of noise step or even try to sample a range of noise steps to make the training more stable. Besides, the function k(t) in the defined function is not described clearly. Is the method sensitive to the choice of function k(t)?

- The method is only tested on a simulated environment within only three scenarios. It is hard to evaluate the method's effectiveness without testing on real-world scenarios.

**Questions:**

- Why is the stick not always in the air if we provide instructions with the verb “jump up” since the rendered images will align more with the states in the air?

- The results with the velocity metric is a bit confusing. Why the diffusion-based ones are faster than the clip-based one? The reward function is to match the text descriptions, which does not imply the velocity.

- How will the model perform if provided with descriptions like “move forward” and “move backward”. Will they generate different policies?

---

> ### Author Response · Authors · 2023-11-21
> **Response to reviewer gAGE**
>
> We thank Reviewer gAGE for their thorough review.  We seek to address their listed considerations below:
>
> - On comparison with text-conditioned work: we appreciate the reviewer for providing a list of related work.  We would like to mention that these works: LangLfP, Text-Conditioned Decision Transformer, and Hiveformer, all require training on datasets of trajectories that have been labeled with natural language.  Not only are such labeled datasets expensive to create, involving substantial human effort, but the resulting trained policies do not naturally transfer across environments.  Indeed, a labeled dataset of trajectories must be created for each novel environment, and across a large number of behaviors and associated text prompts within the environment.  In contrast, TADPol enables the learning of text-conditioned policies irrespective of visual environment, and without requiring any pretraining dataset of demonstrations or labeling whatsoever.  We do agree with the high-level sentiment of the reviewer, however, that additional comparisons with benchmarks would strengthen the work.  In the updated experiments, the results of which can be found in both the updated manuscript and website, we provide comparisons against CLIP-SimPol (Rocamonde et al. arXiv:2310.12921 [a]), LIV (Ma et al., ICML 2023 [b]), and Text2Reward (Xie et al., arXiv:2309.11489 [c]).  Such approaches leverage large-scale pretrained models to generate dense rewards for novel behaviors conditioned on text, and we apply them out-of-the-box to arbitrary RL environments.  We demonstrate that in the Dog environment from DeepMind Control Suite that TADPol is able to outperform the other methods in learning both goal-achieving behaviors as well as continuous locomotion behaviors.
>
> - On the sensitivity to noise steps:  we agree with the reviewer that the choice of noise step might be sensitive to the performance of TADPol. In the updated experiments for the Dog experiment, we thoroughly investigate design choices surrounding the source noise in Appendix B.3.  We explore how the frequency of re-sampling the source noise affects the behavior: having a global source noise for all episodes results in more stable, CLIP-like behavior, having a source noise resampled for every frame results in more a more unstable reward signal and therefore policy, and having a source noise re-sampled for each episode is currently proposed as a happy medium.  Furthermore, we demonstrate that choice of noise level also affects the resulting policy; too high a noise level (such as using step 750 out of 1000), and the reward signal is no longer meaningfully understanding alignment with the generated frame.  We discover a “sweet spot” for the noise level exists around step 450 out of 1000 that is able to extract meaningful priors from the pretrained diffusion model with respect to the generated frame, while avoiding overcorruption.  We visualize the results of these ablation experiments in the updated website link.
>
> - On the $k(t)$ function: for our experiments we use $k(t) = \frac{1}{2} (1 - \prod_{i \leq t} \alpha_i )^2$; however, we note that in our experiments $k(t)$ can be treated as a constant, since we only evaluate with a constant t (where $t=450$ in our experiments).  However, we include a $k(t)$ term in the TADPol equation to keep it general, allowing it to flexibly change with respect to $t$; in experiments where $t$ is resampled during training, this term provides the option to flexibly adjust the reward computation accordingly.
>
> - On testing on more advanced environments: we take note of the reviewer’s wish to see experimentation on more advanced environments.  We therefore report additional experiments on the Dog locomotion environment from the DeepMind Control Suite.  The Dog environment is complex in that it has a high degree of freedom of motion, with no task-specific design decisions, allowing the modeling of very flexible and complex behaviors.  We demonstrate that TADPol is able to learn goal-achieving behaviors (such as standing on its hind legs or chasing its tail), as well as continuous locomotion movements (such as walking) conditioned on natural language inputs, and invite the reviewer to look at the updated website for the latest visualization results.  The background visual environment of the Dog has been modified to appear more natural, and we take these promising results as positive signals for TADPol’s effectiveness in real-world scenarios.

---

> > ### Author Response · Authors · 2023-11-21
> > **Response to reviewer gAGE (cont.)**
> >
> > - On understanding the resulting policy: the reviewer wonders why the stick figure is not always in the air, if the instructions are "jump up", as intuitively rendered images of the stick figure in the air would align more with the text prompt.  We believe that this is avoided due to TADPol using a generative model (namely, a diffusion model) as the supervisory signal.  The large-scale pre-trained diffusion model has learned a distribution over images that align with a particular text prompt, which includes the stick figure at various heights in the air, or even on the ground but preparing to "jump up".  Which particular images in the distribution are most preferred at any particular time is determined by the source noise, which is re-sampled at each episode.  Therefore, TADPol is able to learn more coherent and complete motions that involve a variety of related poses to the text prompt through the learned priors of the pretrained diffusion model.  On the other hand, as elaborated upon in Section 4.2, and Appendix A.1, CLIP-based approaches are deterministic and therefore seek to achieve and maintain a singular frame that provides the highest alignment score.  This results in behavior closer to the hypothetical situation the reviewer posed; however, these approaches are thus naturally inconducive to supervising the learning of continuous locomotion capabilities with no canonical "goal-state" (such as dancing or walking), which we demonstrate TADPol can handle.
> >
> > - On the velocity metric:  we agree that the reward function seeks to match the text descriptions; however, when the text description involves motion (such as in the provided experiment "a stick figure walking across a grid floor"), it is worth evaluating ground-truth motion metrics such as velocity to gauge the level of motion achieved.  By visualizing the video produced by CLIP-SimPol, we qualitatively observe that the resulting agent does not move much from its initial position.  The purpose of the velocity and distance metrics is to quantitatively ground this observation, and explicitly show that CLIP-SimPol prefers to remain stationary in a position that has high alignment with the text prompt.  In comparison, TADPol achieves a higher total velocity and distance achieved, indicating successful motion and therefore higher alignment with the desired locomotion prompt of "a stick figure walking across a grid floor".  Further analysis into why CLIP-SimPol behaves in such a way and why TADPol potentially avoids this pitfall is explored in Section 4.2 and Appendix A.1.
> >
> > - On directional descriptions: as mentioned in Section 4.3 and Appendix A.2, a natural limitation of generating the reward signal solely from individual frames’ alignment with natural language is that there is no conceptual way to determine temporal-extended properties such as direction or speed.  Therefore, we do not expect vanilla TADPol to successfully learn different policies for "walking forward" and "walking backwards" at a conceptual level when such behaviors appear symmetric apart from direction.  However, in this work we propose two potential ways to address this problem: firstly, through a text-to-video diffusion model.  Conceptually, using text-to-video models under the Vid-TADPol framework is promising in that the video generated by the rolled-out trajectory can be evaluated with respect to distance and speed on top of alignment with a text prompt.  However, despite its potential, we discover that current text-to-video models do not outperform text-to-image models under the TADPol framework, most likely due to the quality of current text-to-video models themselves, which is an interesting future direction to explore.  Furthermore, as stated in Appendix A.2, we are aware that it is often desirable to learn policies that consider non-visual factors as well, such as minimizing control costs or the energy expended by the agent as it interacts.  Such information may not be naturally extractable from visual signals, even if videos are used.  In such cases, we believe TADPol can be augmented with other available reward signals that consider other factors beyond the visual domain.  In our updated rebuttal experiments, explained in the new Appendix B.4 section, we demonstrate that TADPol can be augmented easily and effectively - when adding a simple direction-agnostic speed reward to the TADPol reward for the prompt of “a dog walking”, the resulting learned Dog agent indeed is able to move quickly while moving in a natural walking motion.  We believe that a direction term can be a cheap augmentation on top of vanilla TADPol to guide it to “move forward” or “move backward” at will.  Further investigative efforts into how TADPol can be made aware of and utilize motion, temporal, and non-visual information is interesting future work.

---

### Official Review · Reviewer_fbDb · 2023-11-01

**Soundness:** 3 good
**Presentation:** 2 fair
**Contribution:** 3 good
**Rating:** 5
**Confidence:** 3

**Summary:**

This work proposes TextAware Diffusion Policies (TADPols) to leverage  text-to-image diffusion models to generate the reward signals for RL policies, which is from the prediction error between the diffusion model and the rendered images. The experiments show that the TADPols have comparable performances to the baselines with original rewards in some tasks.

**Strengths:**

The method is novel and interesting.

**Weaknesses:**

(1) The writing of this paper requires improvement in a great deal.

(2) The experimental setting is not sufficient.
- There are few baselines for introducing diffusion models for policy learning and other methods (not diffusions) for reward generations.
- The tasks are simple. What about some tasks with the DMControl suite, as other works do.

(3) The reward signal lacks of motion information or temporal information. For example, the diffusion model can not distinguish whether the walker is walking forward or backward. How to identify rewards in these scenarios?

**Questions:**

(1) Because the image is generated by the frozen diffusion model, it must be blurry and quite different from the current scene. In this case, the reward noise will be large, and RL is sensitive to the rewards. So I am wondering why the generated rewards can share comparable performances to the vanilla ones. Can you do some visualization to explain this phenomenon?

(2) The inference of the diffusion model is at a quite low speed. I am wondering how computation-efficient of this work is. Is it necessary to generate rewards every step?

---

> ### Author Response · Authors · 2023-11-21
> **Response to reviewer fbDb**
>
> We thank Reviewer fbDb for their review, and seek to address their listed questions and feedback:
>
> - On writing: we appreciate the reviewer’s encouragement to improve the writing of this paper.  We update our draft to demonstrate and analyze new experiments to substantially reinforce the merits of our proposed approach.  We look forward to the reviewer’s helpful comments on the latest version of the work to improve the clarity of our paper.
>
> - On baseline experiments: In our updated experiments (Section B in appendix) on the Dog environment, we evaluate TADPol against numerous other reward generation methods.  We compare against CLIP-SimPol (Rocamonde et al. arXiv:2310.12921 [a]), LIV (Ma et al., ICML 2023 [b]), and Text2Reward (Xie et al., arXiv:2309.11489 [c]), and demonstrate that TADPol is able to achieve superior performance in not only goal-achieving behaviors but also continuous locomotion.
>
> - On the simplicity of tasks: we agree with the reviewer that the merits of TADPol would be better showcased in complex environments such as DMControl.  We therefore provide additional experimental results on the Dog environment, a complex, flexible continuous control environment from the Deepmind Control Suite.  This environment is complex in that it has a high degree of freedom of motion, and no task-specific reset conditions, allowing the modeling of very flexible and complex behaviors.  We further split our analysis across two types of tasks: goal-achievement and continuous locomotion.  For goal-achieving tasks, we demonstrate that TADPol learns to successfully achieve a desired pose, such as "standing", "standing on hind legs", or "chasing its tail".  For continuous locomotion, we show how TADPol outperforms other approaches both qualitatively and quantitatively in learning motion.
>
> - On motion and temporal information: as mentioned in Section 4.3 we agree with the reviewer that by generating the reward signal solely from individual frames’ alignment with natural language, a fundamental limitation is that there is indeed no conceptual way to determine temporal-extended properties such as direction or speed.  In this work we propose two ways to address this problem; firstly, by proposing and comparing against a text-to-video diffusion model.  Text-to-video models would be able to distinguish if the entire rolled-out trajectory indeed moves according to a desired direction or speed specification.  However, we discover that current text-to-video models do not outperform text-to-image models under the TADPol framework, which is an interesting future direction to explore.  Secondly, as stated in Appendix A.2, we are aware that it is often desirable to learn policies that consider non-visual factors as well, such as minimizing control costs or the energy expended by the agent as it interacts.  Such information may not be naturally extractable from visual signals, even if videos are used.  In such cases, we believe TADPol can be augmented with other available reward signals that consider other factors beyond the visual domain.  In our updated rebuttal experiments, explained in the new Appendix B.4 section, we demonstrate that TADPol can be augmented easily and effectively - when adding a simple direction-agnostic speed reward to the TADPol reward for the prompt of "a dog walking", the resulting learned Dog agent indeed is able to move quickly while moving in a natural walking motion.  We believe further investigative efforts into how TADPol can be made aware of and utilize motion, temporal, and non-visual information is interesting future work.

---

> > ### Author Response · Authors · 2023-11-21
> > **Response to reviewer fbDb (cont.)**
> >
> > - On the reward noise: in TADPol, we do not generate a completely clean image from the frozen diffusion model.  Instead, the procedure corrupts the current scene with a Gaussian source noise, and seeks to re-predict the source noise (or analogously, the clean image) through the diffusion model, conditioned on the text prompt.  The insight of TADPol stems from the fact that if the text prompt is highly aligned with the clean image of the current scene as judged by the diffusion model, then the noise prediction will be highly accurate and the reward for the policy will be large.  On the contrary, if the image is un-aligned with the text prompt, then the text-conditioned denoising operation will struggle to predict the source noise (or reconstruct the current scene), and a low reward will be delivered to the policy.  Furthermore, all aggregated error between the prediction from the text-conditioned diffusion model and the ground truth source noise is converted into a reward signal that is always bounded between 0 and 1, by the equation provided in Section 3.  Therefore, the reward is always a stable value, mitigating RL’s natural sensitivity to reward values.
> >
> > - On inference speed: whereas traditional, vanilla diffusion modeling is slow in inference speed due to the sequential denoising procedure, in TADPol we do not generate the completely clean image for reward computation.  Instead, we only utilize one denoising step to compute a reward for each timestep.  Because we do not perform any iterative procedures for reward computation, and only utilize pretrained models for inference, the reward computation is actually relatively fast.  As such, we are able to train our policy with dense rewards in reasonable wall-clock time; whereas it is interesting to consider generating sparse rewards through TADPol, we operated off the simple assumption that dense rewards are generally preferable.  A study on the merits of dense rewards vs sparse rewards as a design decision is interesting to the RL community at large, and is not limited to TADPol, and TADPol is certainly able to be easily converted into a sparse reward signal to reflect the results of such investigations.

---

### Official Review · Reviewer_mJca · 2023-11-01

**Soundness:** 2 fair
**Presentation:** 1 poor
**Contribution:** 2 fair
**Rating:** 5
**Confidence:** 3

**Summary:**

While previously text-to-video models are trained using text-video pairs. This paper instead proposes to do text-to-video generation, using an existing physics simulator. This helps in preventing the text-to-video model from modelling low-level pixels, physics etc.
Instead now the model learns on how to act in the physics simulator, given a description of the task. The model proposes to use text-to-image diffusion models as reward signals to learn a policy that can act in the physics simulator such that it can generate a video of the text description.

**Strengths:**

- Proposes a novel way of rendering videos.
- Uses a generative model for getting a reward function to learn new behaviours, to my knowledge previous works have mainly use discriminative models such as CLIP for this.
- Shows results indicating that on a some environments they achieve rewards similar to the ground truth reward.

**Weaknesses:**

- the motivation of using simulator to render videos is unclear to me. Like in what realistic scenarios would such a method be useful? As to the best of my knowledge current simulators are not realistic in terms of the RGB they render aka sim2real gap. It's unclear to me in what end use cases would they be useful, also given that we have millions of videos available on the web widely.

- there are no comparisions with existing video rendering methods, thus making it unclear in what cases would they get better results. This point  is linked with the above point.

- The other motivation of the paper is learning robot behaviours, however there are mainly approaches previously proposed that do so such as : https://arxiv.org/abs/2310.12921 https://arxiv.org/abs/2203.12601 https://arxiv.org/abs/2210.00030. The paper fails to compare against any of them.

**Questions:**

Any answers to address the three points above would help me make the final decision.

---

> ### Author Response · Authors · 2023-11-21
> **Response to reviewer mJca**
>
> We thank Reviewer mJca for supplying useful feedback in response to our work, as well as relevant prior work, which we address:
>
> - On simulator-rendered videos: we agree with the reviewer that there are limited realistic scenarios that would utilize the output of a simulator-rendered video as a meaningful video, beyond some complex behavior demonstrations that might be too expensive or dangerous to perform and record in real life (e.g. how to evacuate a failing spacecraft, or how to perform a high-stakes surgical procedure).  We would like to clarify the motivation of using a simulator to render videos as a natural way to treat policies as a form of visual generative model.  By interpreting the policy as a model that can generate coherent images/frames (rendered through an environment), we can connect it with diffusion models, which also generate images but with additional text-conditioning capabilities.  TADPol can then be motivated as a way to distill these capabilities, acquired from large-scale data and architecture, from a pretrained diffusion model into a policy, to naturally unlock the learning of text-conditioned behavior.  We do so by training the policy to generate rendered frames that align with the text prompt as supervised by the priors of a pre-trained frozen diffusion model.  Using a simulator to render videos of a policy is therefore the motivating bridge that connects policy learning with existing large-scale pretrained models, and enables the learning of novel text-conditioned behaviors.
>
> - On comparisons with existing video rendering methods: tied with the previous response on the practical usage of simulator-rendered videos, we agree that our goal is not really to render high-fidelity, natural-looking videos for downstream purposes.  Indeed, the visual quality of the policy is naturally limited by the strength of the environmental renderer.  Aware of these limitations, and as it is also not our main objective, we therefore do not compare TADPol against video rendering methods.  Instead, we believe the contribution of highlighting the inherent video-rendering capabilities of a policy in an environment is valuable as a paradigm shift; it exposes a natural bridge through which benefits of scale can be introduced to reinforcement learning, which has generally been siloed off from the scaling benefits observed in other domains such as vision and language.  In this work we demonstrate the benefits of incorporating large-scale data pretraining and architectures into reinforcement learning through TADPol, unlocking the learning of policies that are flexibly and accurately conditioned on natural language inputs.

---

> > ### Author Response · Authors · 2023-11-21
> > **Response to reviewer mJca (cont.)**
> >
> > - On prior works for learning robot behaviors:  We thank the reviewer for providing a list of previous approaches that learn robot behaviors, but highlight that we already compare against one of the listed works.
> >     - We would like to note that the first work listed is a concurrent work submitted to this year's ICLR, and was made public on arXiv after the submission deadline of this conference; thus there was no way to directly reference its work when writing this submission.  We not only agree that their work is directly relevant to our goals, we in fact do perform comparisons against their proposed approach.  In that work, they propose using a pre-trained, frozen CLIP model to provide a dense reward signal aligning visually rendered observations and text conditioning.  Independently and concurrently, we also created and implemented this approach, termed in our paper as CLIP-SimPol, and compared against it in all of our provided experiments.  We demonstrate that TADPol outperforms CLIP-SimPol, particularly in continuous control tasks such as locomotion, and further provide analysis and justification as to why this is the case (Section 4.2, and Appendix A.1).  We will make sure to update our references section to include their concurrent work.  In our new experiments on the Dog environment, prepared for the rebuttal, we further demonstrate how CLIP-SimPol underperforms in goal-achievement and continuous locomotion compared to TADPol.  The submission link, reviews, and comments for their work are publicly available on OpenReview as well.
> >     - The second work, R3M, is limited to imitation learning (specifically behavior cloning), and was not applied to reinforcement learning settings.  Therefore, the task is to imitate a performed behavior rather than learn a completely novel behavior directly from text specification, which TADPol seeks to do.  Furthermore, not only does R3M require demonstrations of the desired behavior, which TADPol (and other methods we seek to evaluate against) do not, R3M also requires a substantial pretraining dataset with human-labeled text annotations.  This requirement limits R3M’s application to novel environments, in the absence of an expensive human-labeling operation, whereas TADPol can be flexibly applied across many environments out of the box.
> >     - The last listed work, VIP, does not enable the learning of text-conditioned behavior.  Instead, tasks are specified through goal images.  In contrast, in this work we are interested in learning novel, text-conditioned behaviors, as text is a natural, flexible, and human interface to specify desired policy behavior.  On the other hand, we highlight LIV, a follow-up work to VIP, as a way to generate dense rewards from text specification.  In our additional rebuttal experiments on the Dog environment, we include comparisons against LIV, and both qualitatively and quantitatively show that it underperforms on both goal-achieving and continuous locomotion tasks compared to TADPol.  It is noteworthy that both VIP and LIV require pre-training; in particular, LIV requires text-annotated pretraining (they use EpicKitchen, a text-annotated video dataset).  On the other hand, TADPol can be applied to any arbitrary reinforcement learning environment with visual rendering capabilities out of the box, without any prior trajectories or human labeling (which is an expensive procedure).
> >
> > - Additional note: we include, in our rebuttal experiments, an additional comparison against Text2Reward, where a reward function is generated entirely via a LLM.  This reward function has access to ground-truth state attributes, such as speed, direction, and forces on each joint.  However, we demonstrate both quantitatively and qualitatively in the updated manuscript and website that TADPol is able to outperform it in both goal-achieving and continuous locomotion tasks on the Dog environment.

---

### Official Review · Reviewer_LDs4 · 2023-11-04

**Soundness:** 1 poor
**Presentation:** 3 good
**Contribution:** 2 fair
**Rating:** 6
**Confidence:** 4

**Summary:**

The authors present TADPol, a method that leverages a pretrained text-to-image diffusion model to provide a dense reward signal for control tasks based on only a text prompt. They demonstrate success on several locomotion tasks, show improved performance over a CLIP-based baseline, and also evaluate a video-diffusion-based version of their method.

**Strengths:**

The paper is overall well-written, clearly presented, and very easy to follow. The idea of using a diffusion model to provide a dense reward signal is interesting and novel (as far as I know). On the environments that are tested, the method is successful and performs better than CLIP, which is a natural baseline. The result of successfully performing locomotion tasks in zero-shot from text prompts is impressive.

**Weaknesses:**

While the idea of the paper is sound and interesting, I think the experiments are insufficient to demonstrate the efficacy of the method, particularly due to the lack of baselines.

- The experiments are overall very thin. The tested tasks are very simple and not very numerous, which does not really convince me that TADPol is generally applicable or works consistently.
- Baselines are lacking, with a CLIP-based reward being the only baseline. There are other methods use various pretrained models to provide a dense reward: e.g., LIV, which is mentioned in the related works section. Comparison to some other reward learning methods would be appropriate.
- While the paper is overall fairly clear, I feel that the title is a bit of a misnomer and the narrative constructed at the beginning of the paper caused some confusion. The method does not involve a diffusion policy at all; it leverages a text-to-image diffusion model, but only to provide rewards in a policy-agnostic way. I also found all of the talk about the policy as an implicit video-generating model distracting, as well as the discussion of video-generating diffusion models in the related work. I don't think these are fundamentally related to TADPol, since the end goal of the method has nothing to do with generative modeling, but is instead just about achieving good performance in traditional RL tasks. I really think the authors should edit the paper to make sure it does not claim to perform any sort of video synthesis since this is a gross overstatement.

**Questions:**

None

---

> ### Author Response · Authors · 2023-11-21
> **Response to reviewer LDs4**
>
> We thank Reviewer LDs4 for the in-depth, constructive feedback.  Below we seek to address the reviewer’s concerns:
>
> - On experimentation: we take note of the reviewer’s wish to see more thorough experimentation across more complex tasks, and present additional results on the Dog locomotion environment from the DeepMind Control Suite.  The Dog environment is complex in that it has a high degree of freedom of motion, and no task-specific reset conditions, allowing the modeling of very flexible and complex behaviors.  We demonstrate that TADPol is able to learn goal-achieving behaviors (such as standing on its hind legs or chasing its tail), as well as continuous locomotion movements (such as walking) conditioned on natural language inputs, and invite the reviewer to look at the updated website for the latest visualization results.
>
> - On comparison with existing benchmarks: we perform and report additional comparisons against existing approaches that seek to generate dense rewards from natural language specification.  In particular, we benchmark our approach against LIV (Ma et al., ICML 2023 [b]), as well as Text2Reward (Xie et al., arXiv:2309.11489 [c]).  We demonstrated that using a pre-trained LIV out of the box as a dense reward signal is unable to learn meaningful goal-reaching or continuous locomotion capabilities on the Dog agent.  Whereas this may be expected, given that the experiments LIV is trained for are goal-reaching robotics demonstrations, fine-tuning LIV for the Dog task requires explicit ground-truth text-labeling of trajectories, which TADPol does not require.  Furthermore, we report comparisons against a reward function generated from text descriptions, as proposed by Text2Reward.  In these experiments, we prompt ChatGPT with a similar template as proposed in Text2Reward for Hopper, with the details of the particular Dog agent specified instead of the Hopper agent.  We discover that the reward signal generated purely from text and code conditioning is indeed able to promote related behavior in the Dog agent, but at a lower quantitative and quality compared to TADPol despite having access to real-time state signals such as speed and direction.
>
> - On the usage of "diffusion policy": the reviewer is correct that the policy does not synthesize predicted actions using a diffusion process; however, in this work we call our framework a "Text-Aware Diffusion Policy" because it is a policy optimized to respect a text-aware diffusion model.  In essence, this policy is a distillation of a pre-trained text-to-image diffusion model, transferring all the priors within the diffusion model relevant to the prompt and visual environment into a policy network.  Just as a diffusion model generates images from text prompts, so too does the resulting policy continually "generate" frames (by selecting actions and rendering the results through the environment) conditioned on the input text, in accordance with the text-to-image diffusion model it is distilled from.  As it is a diffusion model essentially converted into a policy network, we call it a "Text-Aware Diffusion Policy"; however, we are amenable to clarifying this name with an alternative, such as "Text-Aware Diffusion-Distilled Policy", and are open to further suggestions from the reviewer.
>
> - On video generation and reinforcement learning: we agree with the reviewer that the end goal of TADPol is about achieving good performance in traditional RL tasks, and that video synthesis is not the main contribution or story of the work.  However, we would like to defend the perspective and highlight the value in interpreting policies as implicit video-generators through an environment with rendering capabilities.  By treating the policy as a model that can generate coherent images/frames (where visual quality is admittedly at the mercy of the environment), it naturally provides a connection with diffusion models which also generate images but with the additional benefit of having excellent text conditioning properties.  This shared perspective inspires the TADPol framework, where we seek to distill the priors and text-alignment captured within a pre-trained text-to-image diffusion model, such that the resulting policy also generates frames that are now strongly aligned with human-provided text.  Whereas we do not treat policies as video renderers for the sake of using the output videos, it is precisely because we *treat* policies as a form of video generator that provides a clear motivation and intuition into TADPol.  Indeed, it is through this perspective that a path forward to incorporating the benefits of large-scale pretrained data and models into reinforcement learning, which has generally been siloed off from such scale, becomes promising; and in this work we preliminarily demonstrate such benefits, unlocking the learning of policies that are flexibly and accurately conditioned on natural language inputs.

---

> > ### Comment · Reviewer_LDs4 · 2023-11-22
> >
> > Thank you for the update and extensive additional experiments. The dog results are impressive and have convinced me of the method's robustness much more than the original experiments. The addition of LIV and Text2Reward are important baselines that also flesh out the experiments section.
> >
> > Overall, it seems like this paper is still a bit of a work in progress. However, zero-shot text-conditioned reward shaping is a hot area right now and I understand the authors' desire to get something out there. The authors' proposed method is novel, interesting, and appears to work at least as well as competing methods. I also appreciate the authors' attention to detail and ability to produce high-quality work. As a combination of these factors, I am slightly positive about accepting the paper, and have raised my score to a 6.
> >
> > I do think the biggest remaining weakness is the writing structure and narrative, which made the paper much more difficult to digest (for me as well as other reviewers, it seems). The first issue is the title: having anything close to "diffusion policy" immediately evokes the existing idea of [diffusion policies](https://diffusion-policy.cs.columbia.edu/), which are quite popular in the robotics community and completely unrelated to TADPol. The second issue is the framing of the paper, at least in the introduction, as being about "video generation". This immediately evokes video generation work from the generative modeling literature, and sets expectations accordingly, only for them to be crushed when the reader realizes that the "video generator" is a MuJoCo renderer. While I appreciate the authors' perspective on the connection to video synthesis, and this is an interesting point, I think it would be possible to more gently introduce the reader to this perspective without misleading (and potentially upsetting) them.
> >
> > My overall point here is that the authors should be very up-front and transparent about what the paper does at its core, which from my perspective, is primarily about *using a generative diffusion model in a discriminative manner to provide text-conditioned rewards*. When framed this way, I personally still think the idea and results are pretty cool. This also makes it much easier for the reader to categorize the paper and know what comparisons to expect, i.e., comparisons to other reward-shaping papers such as LIV.

---

> > > ### Author Response · Authors · 2023-11-23
> > > **Thank you for your response.**
> > >
> > > We thank Reviewer LDs4 for their comments, especially this late in the discussion period, and are happy to hear that we have addressed many of their concerns.  We appreciate the reviewer raising their score.
> > >
> > > We agree with the reviewer that the current writing structure and narrative may caused other reviewers confusion regarding what our main objective is, and could be improved.  We therefore take the reviewer's suggestion to rewrite the introduction to focus less on video generation, which is not the core of the work, and to instead motivate and highlight our main contribution of text-conditioned policy learning.
> > >
> > > We agree with the reviewer's analysis that TADPol is indeed about using a generative diffusion model in a discriminative manner to provide text-conditioned rewards.  We hope that the reviewer finds our updated introduction, highlighted in blue, to be a more up-front, transparent, and accurate depiction of our work and contributions.

---

### Author Response · Authors · 2023-11-21
**Our general response**

We thank the reviewers for their thorough, thoughtful, and constructive comments.  We are happy to hear that the reviewers consistently agree that adopting diffusion models as a reward model is both novel and interesting.  In particular, the reviewers highlighted that prior works that attempt to learn zero-shot text-conditioned policies use discriminative models such as CLIP, and that TADPol’s approach of using a generative model is fresh and achieves superior results.

A common comment from the reviewers was that additional experimentation on complex tasks and environments, as well as additional baselines would strengthen the work.  We therefore provide the following:

- In our updated manuscript and website, we showcase new results on the Dog locomotion environment from the DeepMind Control Suite.  The Dog environment is complex in that it has a high degree of freedom of motion, and has no task-specific design decisions (such as reset conditions), allowing the flexible modeling of expressive and complex behaviors.  We report the ability of TADPol to learn both goal-achieving tasks, such as striking a particular pose, as well as continuous locomotion tasks.  It should be noted that achieving continuous locomotion from text conditioning is a difficult endeavor that has not been thoroughly explored in prior work, to this author’s knowledge; whereas in goal-achievement a desired final goal state can be created or selected, and the task becomes a form of inversion to achieve it, in continuous locomotion tasks such as "walking", "rolling", or "dancing", there is no canonical pose or frame that, if achieved, represents a completion of the task.  Instead, such activities have to continuously cycle through multiple motions (swinging legs in a coherent sequential motion, for walking) to accurately constitute satisfying the desired behavior.

- We demonstrate and report a more comprehensive set of baselines for the Dog tasks, including CLIP-SimPol [a], LIV [b], and Text2Reward [c].  All of these methods leverage large-scale pretrained models to generate dense rewards for novel behaviors conditioned on text, and we apply them out-of-the-box to arbitrary RL environments such as Dog.  Unlike many other approaches, TADPol and these baselines do not require a dataset of trajectories labeled with natural language captions or instructions, which is expensive to procure and limits generalization of the approach.  Of these baselines we note that Text2Reward, which generates a reward function as the output of a LLM, does not have access to any visual information, but does have access to ground-truth state information for the dog including speed, direction, and joint positions.

In our updated experiments, we demonstrate that TADPol is able to learn goal-achievement behaviors that outperform the other baselines, such as making a dog stand up with the prompt "a dog standing".  Furthermore, TADPol respects subtle variations to the prompt, successfully demonstrated through the prompt “a dog standing on hind legs”.  We also showcase how TADPol is able to outperform other baselines in learning continuous locomotion for the dog, as judged through the prompt "a dog walking".  We further show how TADPol can be cheaply augmented with non-visual signals, such as minimizing energy cost or maximizing speed, to learn a complete policy that respects multiple desiderata simultaneously.  We invite the reviewers to review the updated manuscript, with edits highlighted in blue, as well as the videos on the updated website (https://sites.google.com/view/tadpol-iclr24/). Note that due to the number of long GIFs, the website may need 2-3 minutes to load fully.

We would also like to reiterate the core contribution of our work:

- We propose TADPols, where a frozen, pretrained diffusion model is leveraged to supervise the learning of policies that behave in alignment with natural language.
- We demonstrate that TADPols can learn not only goal-achieving behavior, but also continuous locomotion capabilities that are aligned with text specification.
- We show that TADPol outperforms other approaches (such as using CLIP as a supervisory signal) in learning novel text-conditioned policies, and extends the capabilities of learned RL policies beyond the confines of ground-truth hand-designed reward functions.

We thank the reviewers for their time and careful consideration!

[a] Juan Rocamonde et al. Vision-language models are zero-shot reward models for reinforcement learning. arXiv preprint arXiv:2310.12921, 2023.

[b] Yecheng Jason Ma et al. Liv: Language-image representations and rewards for robotic control. ICML, 2023.

[c] Tianbao Xie et al. Text2reward: Automated dense reward function generation for reinforcement learning. arXiv preprint arXiv:2309.11489, 2023.

---

### Author Response · Authors · 2023-11-23
**A final update**

As the discussion period closes, we thank all reviewers for their helpful comments and feedback on how our work could be improved.  A common suggestion we received was to improve our introduction to focus on our true main objective of learning text-conditioned policies for robot behaviours, rather than highlight the text-to-video interpretation of our work (which is not our core objective, but rather a motivating observation).  Following this advice, we have thus updated the manuscript with a revised introduction (written in blue).  We hope that the reviewers find the new introduction to be a better description of our method's motivations, objectives, and contributions.  Thank you all for your time and consideration.

---

### Meta-Review · Area_Chair_aDTu · 2023-12-06

**Metareview:**

This paper addresses the problem of text based reward learning using a diffusion model. The authors use a DM to provide dense reward signal for RL, which is interesting and novel. On the environments that are tested, the method seems to be successful and perform better than CLIP. The method also performs well on locomotion tasks in zero-shot from text prompts, which is interesting.

One of the main concerns raised by the reviewers is the poor narrative. Many reviewers got confused that this is one of the video generation papers when this was not really the contribution of the paper. While the perspective on connection to video generation is interesting, its important not to confuse the readers. The authors should really work on the narrative.

The other concern is lack of experiments. The reviewers believe that the experiments considered are too simple and the paper needs more solid experiments. The authors included dog locomotion experiments in the rebuttal which was an important one to include. The paper got better with these results, however, some reviewers have concerns about the generalization ability of the method in real-world applications.

At this stage, I believe the paper is not ready for publication. During discussions with the reviewers, we agree that the paper is still in early phase and needs quite a bit of revision to change the narrative and include some more real world experiments. So, I currently vote for rejecting the paper. While this might be disheartening for the authors, I would strongly encourage them to work on the paper and have a strong submission to the next venue.

**Justification For Why Not Higher Score:**

The paper needs improvement in writing and some strong results on real world datasets. The paper is still in early stage and not ready for publication.

**Justification For Why Not Lower Score:**

N/A

---

### Decision · Program_Chairs · 2024-01-16

Reject